# Selective packaging of mitochondrial proteins into extracellular vesicles prevents the release of mitochondrial DAMPs

Kiran Todkar[1,2,3], Lilia Chikhi[1,2], Véronique Desjardins[1,2], Firas El-Mortada[1,2], Geneviève Pépin[1,2,3] & Marc Germain [1,2,3✉]

Most cells constitutively secrete mitochondrial DNA and proteins in extracellular vesicles (EVs). While EVs are small vesicles that transfer material between cells, Mitochondria-Derived Vesicles (MDVs) carry material specifically between mitochondria and other organelles. Mitochondrial content can enhance inflammation under pro-inflammatory conditions, though its role in the absence of inflammation remains elusive. Here, we demonstrate that cells actively prevent the packaging of pro-inflammatory, oxidized mitochondrial proteins that would act as damage-associated molecular patterns (DAMPs) into EVs. Importantly, we find that the distinction between material to be included into EVs and damaged mitochondrial content to be excluded is dependent on selective targeting to one of two distinct MDV pathways. We show that Optic Atrophy 1 (OPA1) and sorting nexin 9 (Snx9)-dependent MDVs are required to target mitochondrial proteins to EVs, while the Parkinson's disease-related protein Parkin blocks this process by directing damaged mitochondrial content to lysosomes. Our results provide insight into the interplay between mitochondrial quality control mechanisms and mitochondria-driven immune responses.

[1] Groupe de Recherche en Signalisation Cellulaire and Département de Biologie Médicale, Université du Québec à Trois-Rivières, Trois-Rivières Quebec, Canada. [2] CERMO-FC UQAM, Quebec, Canada. [3] Réseau Intersectoriel de Recherche en Santé de l'université du Québec, Université du Québec, Québec, Canada. ✉email: marc.germain1@uqtr.ca

Most cells secrete a range of extracellular vesicles (EVs) that act as communication devices by carrying proteins and nucleic acids between cells[1–3]. EVs can be broadly divided in two classes: exosomes, small vesicles arising from the fusion of a multivesicular body with the plasma membrane, and the larger microvesicles, which are thought to arise from the direct budding of vesicles from the plasma membrane[4]. Intriguingly, a number of studies have identified mitochondrial proteins as EV cargo[5–10].

Mitochondria are essential organelles that act as a central metabolic hub. As a consequence, mitochondria regulate a number of key cellular processes, ranging from the production of cellular energy to the induction of apoptosis and cellular differentiation[11–13]. Impaired mitochondrial function has thus major impacts on both cells and the organism, which are compounded by the production of toxic reactive oxygen species (ROS) by damaged mitochondria[14,15]. Importantly, damaged mitochondria, or the release of N-formyl peptides and mitochondrial DNA from mitochondria can act as damage-associated molecular patterns (DAMPs) that activate the innate immune system[16–22]. In fact, mitochondrial cargo within EVs and free mitochondria released by some cell types following pro-inflammatory stimulation like exposure to lipopolysaccharide (LPS) have been shown to stimulate the production of proinflammatory cytokines, further enhancing LPS-induced inflammation[21,23–27].

While inflammation promotes the secretion of mitochondrial content, mitochondrial proteins are also clearly present in EVs under unstimulated conditions[9,28]. The mechanism by which they are secreted and whether they also participate in immune activation remains unknown. Here we show that cells selectively target damaged mitochondrial components for lysosomal degradation to prevent the release of this pro-inflammatory content into EVs. This process is dependent on mitochondria-derived vesicles (MDVs), small vesicles that carry mitochondrial proteins to other organelles. Specifically, this accurate sorting of mitochondrial components requires two distinct MDV pathways. First, delivery of mitochondrial proteins to EVs requires Snx9-dependent MDVs, a subset of MDVs that were previously shown to regulate mitochondrial antigen presentation[29]. Second, MDVs carrying damaged mitochondrial components are instead targeted for lysosomal degradation in a process that depends on the Parkinson's disease-related protein Parkin. Altogether, our results demonstrate that cells selectively regulate the packaging of mitochondrial protein into EVs to prevent the release of damaged components that would otherwise act as pro-inflammatory DAMPs.

## Results

**Mitochondria, not EVs stimulate a strong IL6 response.** Mitochondrial content, especially oxidized components, can act as DAMPs that activate an inflammatory response when present in the cytosol or released from cells[16–22]. This can be demonstrated by exposing immune cells, such as the RAW264.7 macrophage cell line, to mitochondria isolated from other cells (here mouse embryonic fibroblasts (MEFs), Supplementary Fig. 1a) and monitoring two different inflammatory pathways (Interferon using IP10, and NF-κB using IL6). Exposure of RAW cells to isolated mitochondria caused a dramatic, dose-dependent increase in IL6 (Fig. 1a) but no significant change in IP10 (Fig. 1b), as measured in the cell culture media. Mitochondrial content could readily be found in EVs isolated from different cell lines under conditions where cell death was kept below 5% to avoid the presence of apoptotic bodies (Protein yield in Supplementary Fig. 1B; Cell death in Supplementary Fig. 1C; Mitochondrial content in Fig. 1c), but was absent from

non-conditioned media (Supplementary Fig. 1D). Therefore, we then tested the ability of EVs to stimulate inflammation using EVs isolated from MEFs, as MEFs provided more consistent EV yields than the other non-immune cells we tested. Compared to mitochondria, EVs caused a significant increase in IP10 (Fig. 1b; $10 \times 10^6$ cells yield 120 µg mitochondria and 12 µg EVs.) but did not stimulate IL6 production (Fig. 1a). This suggests that while EVs stimulate IP10 production, they do not activate the strong IL6 response associated with mitochondrial DAMPs.

We then determined whether inducing oxidative damage to mitochondria further stimulated the IP10 response. For this, MEFs were treated for 24 h with Antimycin A (AA), a complex III inhibitor that stimulates the production of ROS[30–32] but causes minimal cell death (Supplementary Fig. 1E). RAW cells were then exposed to EVs or mitochondria collected from these cells. For these experiments, we used an equivalent amount of proteins from mitochondria and EVs (12 µg mitochondria, corresponding to protein yield of EVs isolated from $10 \times 10^6$ cells), limiting the very strong IL6 response to mitochondria while allowing the detection of IP10 responses in EVs. Consistent with oxidized mitochondrial components being more inflammatory, mitochondria isolated from AA-treated cells, but not control mitochondria induced significant IP10 secretion (Fig. 1d). In contrast, EVs isolated from AA-treated cells did not further stimulate IP10 secretion (Fig. 1d), suggesting that these mitochondrial DAMPs are prevented from being incorporated into EVs. In fact, the examination of mitochondrial content revealed that the Complex I subunit NDUFA9 and the Complex III subunit UQCRC2 were selectively degraded following AA treatment (Fig. 1e). In contrast, the matrix proteins mtHSP70 and PHD were not affected (Fig. 1e). Nevertheless, mtHSP70 levels within EVs were significantly decreased (Fig. 1f), while PDH levels were very low in control EVs and further decreased following AA treatment (Fig. 1f). In contrast, the amount of the mitochondrial outer membrane protein TOM20 found in EVs was not affected by AA treatment (Fig. 1f), consistent with outer membrane proteins not being affected by AA treatment[33]. Interestingly, the addition of the antioxidant N-acetyl-cysteine (NAC) to the AA-treated cells rescued the mtHSP70 incorporation into EVs (Supplementary Fig. 1F), supporting the role of oxidative damage in this process. Altogether, these results suggest that cells selectively regulate the incorporation of mitochondrial components within EVs to prevent the secretion of pro-inflammatory DAMPs.

**Mitochondrial proteins are selectively enriched in EVs.** To identify the mechanism through which cells regulate the inclusion of mitochondrial proteins into EVs, we first characterised MEFs EVs and their mitochondrial content. EV preparations were enriched in the exosomal markers Alix and CD9, but excluded the nuclear protein Lamin B1 and the recycling endosome protein Rab11 (Fig. 2a). Protein quantification by normalizing the amount present in EVs to the cellular content confirmed that the exosome marker Alix was enriched in our EV preparations, while different endosomal markers were not (Rab11, Rab9 (late endosome), Snx9 (endocytosis); Fig. 2b), consistent with the selective inclusion of proteins within EVs. Having demonstrated the selectivity of EV content isolated from MEFs, we quantified the presence of a number of mitochondrial proteins in these EVs and found striking differences in the enrichment of individual mitochondrial proteins (Fig. 2b). Among the proteins tested, TOM20 was present at the highest levels in unstimulated EVs, while PDH and the intermembrane space protein Cytochrome c (Cyt C) were present at very low levels (Fig. 2b). Proteins associated with the inner membrane (mtHSP70, NDUFA9 and the mitochondrial

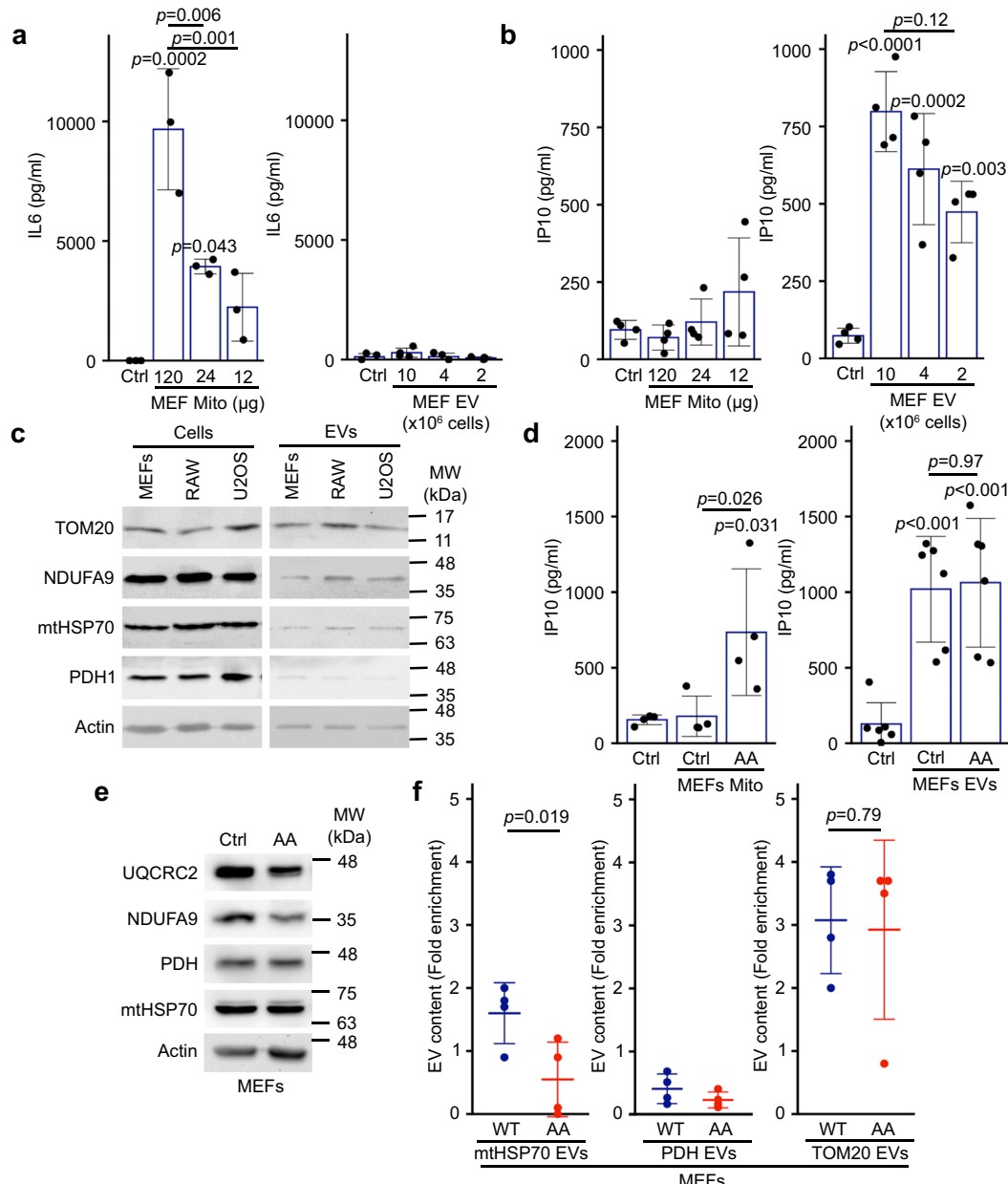

**Fig. 1 Mitochondria and EVs activate distinct pro-inflammatory cytokines. a**, **b** Extracellular mitochondria induce the secretion of pro-inflammatory cytokines. Mitochondria (Mito)(isolated from MEFs) or EVs (isolated by differential centrifugation from the media of MEFs grown for 24 h in EV-depleted media) were added to RAW cells in their culture media. The release of IL6 (**a**) and IP10 (**b**) into culture media was measured by ELISA 24 h. Individual points represent independent experiments (**a**, $n = 3$; **b**, $n = 4$). Bars show the average ± SD. One-way ANOVA. **c** Representative western blot showing the amount of the specified mitochondrial proteins in the indicated cell types (20 μg) vs their EVs (5 μg). TOM20, outer membrane protein; NDUFA9, Complex I subunit; mtHSP70 matrix protein associated with the IM; PDH, matrix protein. Actin is used as a control, as it has been shown to associate with EVs. **d** AA-treated MEFs mitochondria stimulate IP10 production. RAW cells were treated as above with EVs (from $10 \times 10^6$ cells) and mitochondria (12 μg) isolated from Control or AA-treated MEFs, and IP10 release in culture media measured by ELISA. Individual points represent independent experiments (mitochondria, $n = 4$; EVs, $n = 6$). Bars show the average ± SD. One-way ANOVA. **e** AA treatment of WT MEFs causes the selective degradation of mitochondrial proteins. MEFs were treated with AA for 24 h and the indicated mitochondrial proteins analysed by western blot. **f** Enrichment of the indicated mitochondrial proteins in EVs was measured by western blot in Control (blue) and AA-treated (24 h, Red) WT MEFs. Individual points represent independent experiments ($n = 4$). Bars show the average ± SD. Two-sided $t$-test.

fusion protein OPA1) were present at intermediate levels (Fig. 2b). In addition to mitochondrial proteins, mitochondrial DNA (mtDNA) has been reported to be released from different cell types[6,24,34]. Consistent with this, we found that mtDNA was enriched in unstimulated MEF EV preparations (Fig. 2b).

Some proteins are peripherally associated with EVs rather than being sequestered inside these vesicles[35]. We thus tested whether

mitochondrial proteins found in EVs were sensitive to trypsin digestion. As shown in Fig. 2c, mitochondrial proteins were protected from trypsin digestion in EV preparations, unless the membranes were solubilised with Triton X-100 (Fig. 2c; TX100), indicating that mitochondrial proteins are found inside EVs. Overall, the distinct pattern we observed for the inclusion of mitochondrial content within unstimulated EVs suggest that this

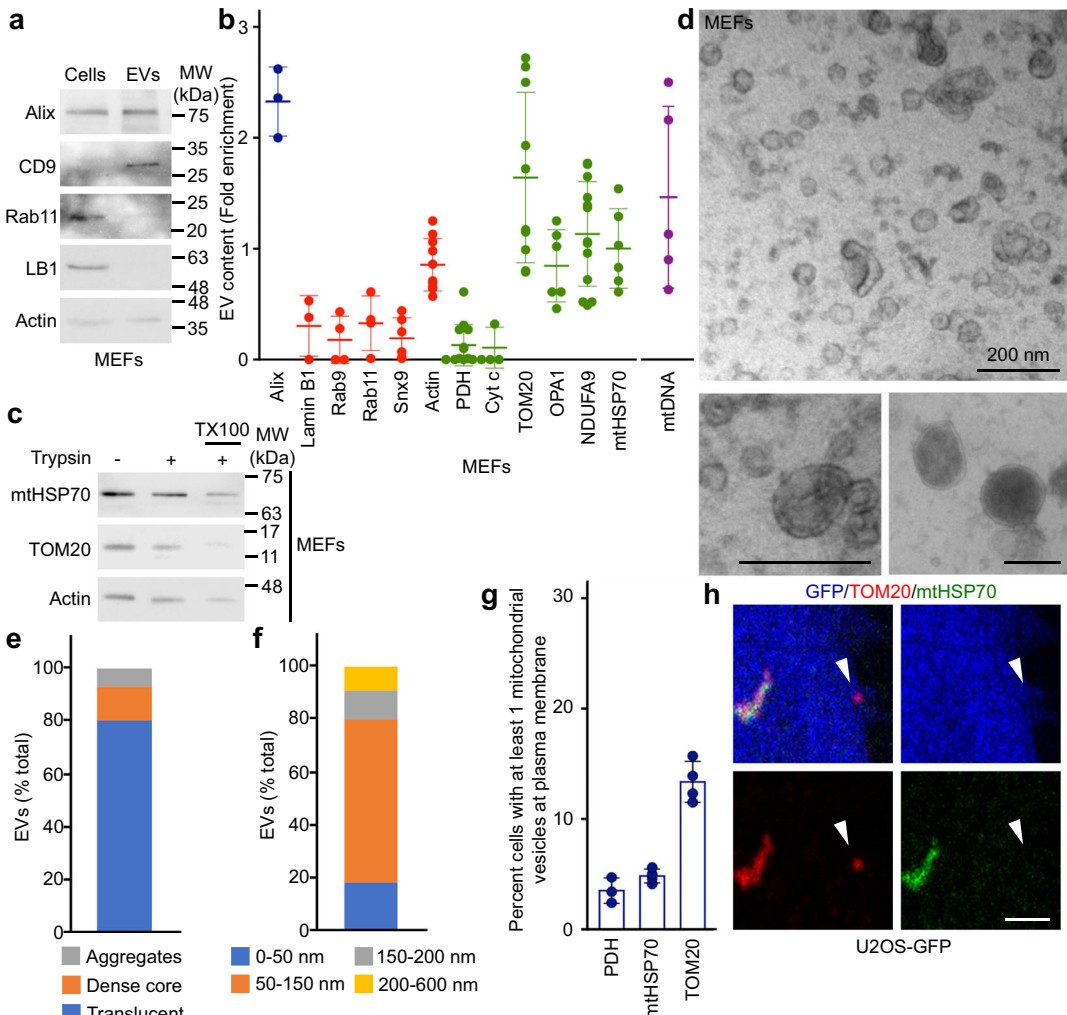

**Fig. 2 Selective inclusion of mitochondrial proteins in EVs. a** EV markers were analysed by western blot in 20 µg cell extracts and 5 µg MEFs EVs isolated as in Fig. 1. **b** Quantification of protein inclusion in MEF EVs. The amount of the indicated protein present in EVs was normalised to its cellular content. Individual points represent independent experiments. Bars show the average ± SD. Alix, Exosome marker (blue); Lamin B1, nuclear protein; Rab9, Rab11, Snx9, endosomal proteins; Cyt c, IMS protein, OPA1, IM protein. Mitochondrial proteins (green), non-mitochondrial proteins (red), mtDNA (purple). **c** MEF EVs isolated as above were treated with Trypsin in the absence or the presence of detergent (TX100) and analysed by western blot for the presence of the indicated proteins. **d**–**f** MEF EV ultrastructure was analysed by EM and quantified based on structure (**e**) and size (**f**). Representative images are shown in (**d**). Scale bars, 200 nm. **g**, **h** Vesicles containing selective mitochondrial cargo are found in proximity to the plasma membrane (≤1 µm) but away from the main mitochondrial network (>1 µm). The number of TOM20-positive, mtHSP70-positive or PDH-positive vesicles are quantified in (**g**) with individual points representing the fraction of cells with plasma membrane-associated vesicles in four independent experiments. Each positive cell typically contained one vesicle. Points show independent experiments and bars show the average ± SD. A representative image is shown in (**h**) with GFP (blue) used as a cytosolic marker to identify cellular boundaries. The arrowhead denotes a TOM20-positive vesicle close to the plasma membrane. Scale bar, 2 µm.

is a selective process distinct from bulk mitochondrial export. To confirm this, we examined our EV preparations by electron microscopy (EM). The majority of the vesicles were small, electron translucent and often cup-shaped (Fig. 2d, bottom left panel, quantified in 2e), consistent with exosomes[28,35]. A second population of vesicles had a distinct structure, containing a dense core (Fig. 2d bottom right panel, quantified in 2e), while a small amount of material seemed aggregated and did not have a clear ultrastructure (Fig. 2d, e). These vesicular structures were not observed in material isolated from non-conditioned media (Supplementary Fig. 1G). Quantification of vesicle size indicated that the majority of the isolated EVs were <150 nm in diameter, while the few larger vesicles were smaller than 600 nm (Fig. 2f). Importantly, we did not find vesicles with a structure or size consistent with whole mitochondria. Altogether, these results are consistent with the reported characteristics of isolated EVs

(including exosomes and microvesicles)[28,35], not extracellular mitochondria.

**Snx9 regulates the inclusion of mitochondrial IM/matrix proteins in EVs**. To determine if this process is associated with the presence of intracellular vesicles accumulating specific mitochondrial proteins, we used U2OS cells, large flat cells that are easy to visualise by confocal microscopy. We stained U2OS cells stably expressing GFP (to mark cytosol) for TOM20 and mtHSP70, two mitochondrial proteins enriched in our EV preparations (Fig. 2b). Structures positive for one mitochondrial marker but not the other were present close to the plasma membrane of these cells (Fig. 2g, h). In addition, there were more TOM20-positive structures than mtHSP70-positive structures (Fig. 2g), correlating with the amount of each protein

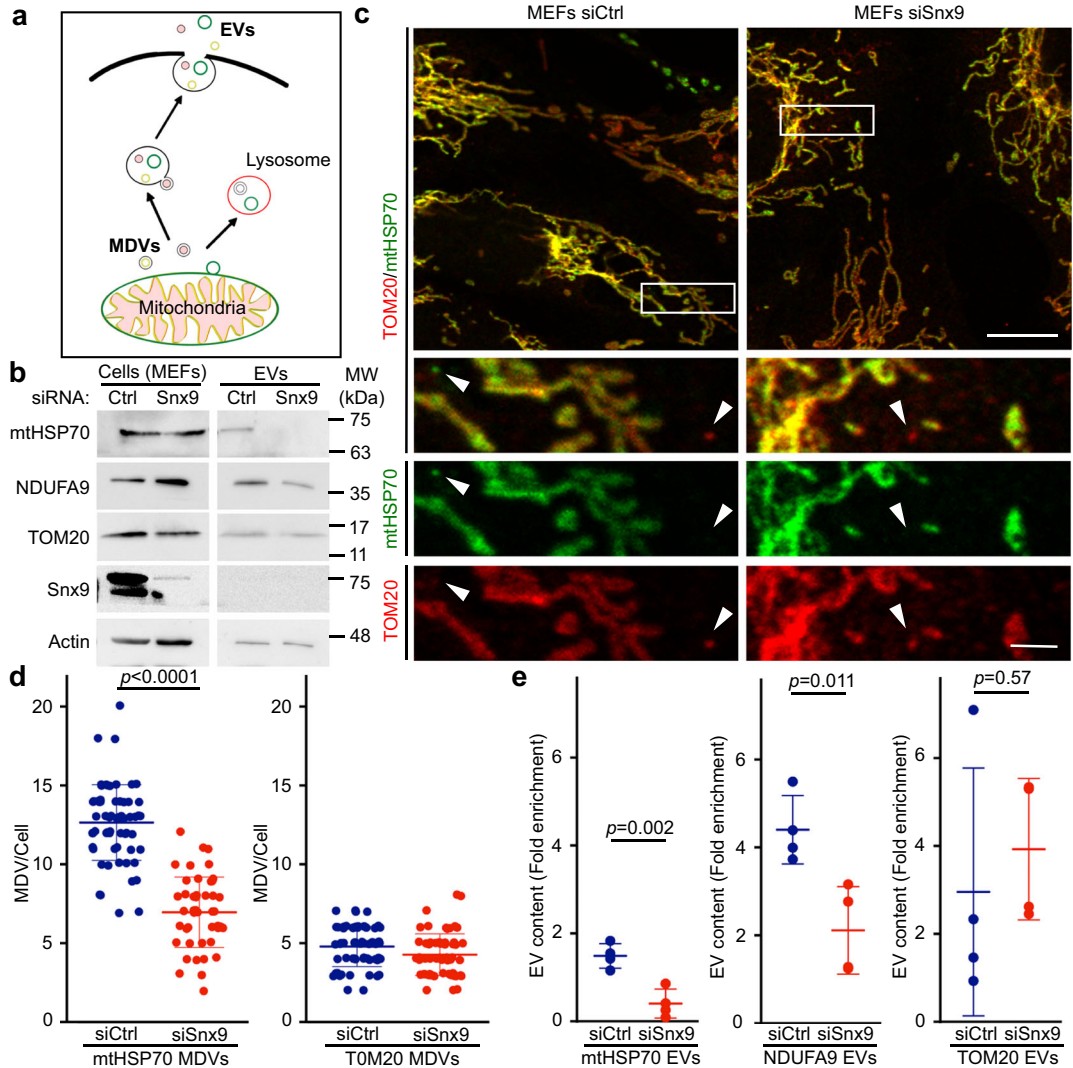

**Fig. 3 Snx9-dependent MDVs contribute to the inclusion of IM/matrix proteins into EVs. a** Schematic representation of the two distinct MDV pathways leading to EVs and lysosomes, respectively. **b** MEFs were treated for 24 h with a control siRNA (siCtrl, blue) or a siRNA against Snx9 (siSnx9, red). Mitochondrial proteins were then measured by western blot, with Actin serving as a loading control. **c** Representative images showing TOM20 and mtHSP70 MDVs in MEFs treated with a control siRNA (siCtrl) or a siRNA against Snx9 (siSnx9). Arrowheads denote MDVs (positive for one marker but negative for the other). Scale bars: top panels, 10 μm; enlargements in bottom panels, 2 μm. **d** MDV quantification from images as in (c). Each data point represents one cell. Bars represent the average of 40 cells in three independent experiments ± SD. Two-sided t-test. **e** Enrichment of the indicated mitochondrial proteins was measured by western blot as in Fig. 1. Individual points represent independent experiments (n = 4). Bars are shown as average ± SD. Two-sided t-test.

present in EVs (Fig. 2b). Similarly, few vesicles positive for PDH (present at low levels in EVs) were present close to the plasma membrane (Fig. 2g). These results are thus consistent with the presence of an intracellular vesicular pathway regulating the selective inclusion of mitochondrial proteins into EVs. The cargo-selective structures we observed are highly reminiscent of MDVs, small cargo-selective vesicles budding off mitochondria that are used to transport specific mitochondrial proteins to other organelles, including lysosomes where the material is degraded[36,37]. As this suggests that MDVs could target mitochondrial proteins to EVs, we addressed their role in EV cargo selection of mitochondrial proteins (general hypothesis in Fig. 3a).

A recent study demonstrated that Snx9, a dynamin-binding protein essential for clathrin-mediated endocytosis, is required for the production of MDVs positive for the mitochondrial matrix

protein PDH[29]. To confirm that Snx9 could be used to manipulate IM/matrix MDV formation, we knocked down Snx9 using siRNA (Fig. 3b, Supplementary Fig. 2A) in MEFs. Snx9 knockdown did not affect intracellular mitochondrial content or overall mitochondrial morphology (Fig. 3b, c, Supplementary Fig. 2A). We thus measured the presence of MDVs in these cells. MDVs are defined by their cargo selectivity, and distinct mechanisms are thought to regulate the formation of MDVs containing outer membrane proteins and IM/matrix proteins, the latter forming double-membrane vesicles lacking outer membrane markers[36,37]. We thus labelled cells with TOM20 (outer membrane) and mtHSP70 (matrix) and counted the total number of small vesicles positive for one marker but not the other (Fig. 3c). Loss of Snx9 caused a significant reduction in MDVs positive for mtHSP70 and negative for TOM20 (mtHSP70 MDVs), but did not affect MDVs positive for TOM20 and negative for mtHSP70 (TOM20 MDVs) (Fig. 3d). This indicates that Snx9

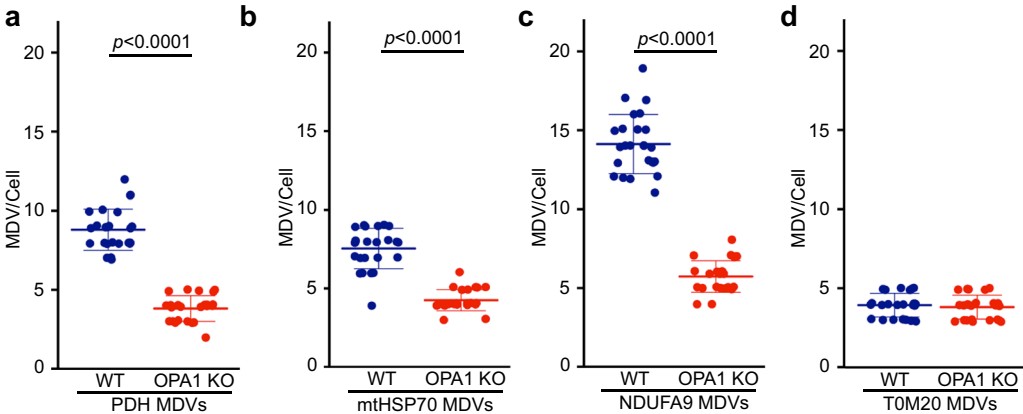

**Fig. 4 OPA1 deletion inhibits the formation of IM/matrix MDVs.** Quantification of MDVs positive for PDH (**a**), mtHSP70 (**b**), NDUFA9 (TOM20-negative) (**c**), as well as TOM20-positive (mtHSP70-negative) (**d**) in immunofluorescence images of WT (blue) and OPA1 KO (red) MEFs. Each data point represents one cell. Bars represent the average of 23 cells (except for PDH where $n = 20$) in three independent experiments ± SD. Two-sided $t$-test.

regulates the formation of IM/matrix MDVs but not outer membrane MDVs.

Having confirmed that Snx9 knockdown can be used as a tool to define the role of MDVs in the inclusion of IM/matrix proteins into EVs, we isolated EVs from siCtrl and siSnx9 cells media and measured their mitochondrial EV content. Consistent with the hypothesis that MDVs are required to target mitochondrial proteins to EVs, mtHSP70 and NDUFA9 were dramatically reduced in siSnx9 EVs (Fig. 3e, Supplementary Fig. 2B). In contrast, but consistent with the MDV data (Fig. 3d), knockdown of Snx9 did not affect the cellular levels of TOM20 (Fig. 3b, Supplementary Fig. 2B) or decrease its inclusion into EVs (Fig. 3e, Supplementary Fig. 2B). The selective loss of MDV formation and EV incorporation of mitochondrial proteins upon Snx9 knockdown thus suggest that MDVs regulate the inclusion into EVs of mitochondrial proteins associated with the mitochondrial IM/matrix.

**Loss of OPA1 inhibits MDV formation and inclusion of mitochondrial proteins in EVs.** To confirm our hypothesis, we sought for an independent way to prevent MDV formation without directly affecting the endosomal compartment. OPA1 is a mitochondrial IM protein required for mitochondrial IM fusion and maintenance of cristae structure[38–41]. Given that OPA1 was one of the mitochondrial proteins enriched in our EV preparation (Fig. 2b) and that its dynamin-like activity could promote the budding of IM/matrix MDVs from the IM, we reasoned that genetic deletion of OPA1 should abrogate the formation of IM/matrix-derived MDVs, but not of outer membrane MDVs. To determine whether this was the case, we measure MDV formation in WT and OPA1 KO MEFs (Fig. 4, representative images shown in Supplementary Fig. 3).

We first measured PDH MDVs (PDH-positive, TOM20-negative), as these MDVs have been extensively studied[29,36,42,43]. As shown in Fig. 4a, the number of PDH MDVs was greatly reduced in OPA1 KO MEFs. NDUFA9 and mtHSP70 MDVs were similarly reduced in OPA1 KO MEFs (Fig. 4b, c), further supporting a role for OPA1 in the generation of IM/matrix MDVs. Importantly, TOM20 MDVs were not affected by OPA1 deletion (Fig. 4d).

As these results demonstrate that OPA1 deletion selectively affects the formation of MDVs containing IM/matrix proteins, we used OPA1 KO cells to test our hypothesis that MDVs are required to package IM/matrix proteins into EVs. Size, overall

protein content and Alix expression were similar between WT and OPA1 KO EVs (Fig. 5a, b; Supplementary Fig. 2C), and these contained similar amounts of cytoplasmic proteins (Fig. 5c, ACC1 and DJ1 quantified in 5d). As these results indicate that overall EV formation is not disrupted by the loss of OPA1, we measured the incorporation of mitochondrial proteins in WT and OPA1 KO EVs. At the cellular level, none of the mitochondrial proteins examined were affected by OPA1 deletion (Fig. 5e). However, consistent with MDVs being required for the inclusion of IM/matrix proteins into EVs, NDUFA9 and mtHSP70 were almost completely absent from OPA1 KO EVs (Fig. 5e, f). Other electron transport chain components (SDHA (Complex II), UQCRC2 (Complex III) were also affected (Fig. 5e). In contrast, TOM20 was still incorporated into EVs in OPA1 KO MEFs (Fig. 5e, quantification in 5f), further supporting the selectivity of this process. Importantly, mitochondrial EV content was rescued by re-expressing OPA1 in OPA1 KO cells (Supplementary Fig. 2D–F).

As mtDNA was present in WT EVs, we also determined the effect of OPA1 deletion on mtDNA incorporation into EVs. Similar to IM/matrix proteins, mtDNA content was decreased in EVs isolated from OPA1 KO MEFs (Fig. 5g; relative to cellular levels to take into account mtDNA differences between WT and OPA1 KO MEFs[38]). Altogether, our results indicate that OPA1 regulates the formation of IM/matrix MDVs, which are then required for the selective inclusion of mitochondrial content into EVs.

**OPA1 KO mitochondria, but not EVs enhance IP10 secretion.** Overall, our results indicate that MDVs are required to package IM/matrix proteins into EVs. As we observed that increased mitochondrial ROS (AA treatment) prevents the inclusion of mitochondrial proteins into EVs (Fig. 1f, Supplementary Fig. 1F), our results suggest that this pathway is inhibited following oxidative damage to mitochondria. OPA1 deletion prevents MDV formation and packaging of mitochondrial proteins into EVs, but also leads to oxidative damage to mitochondria[30,44], allowing us to directly test if inhibiting MDV formation alters the release of pro-inflammatory cytokines caused by oxidized mitochondrial components. We thus measured the response of RAW cells to mitochondria and EVs isolated from WT and OPA1 KO cells. Consistent with our previous results (Fig. 1), IP10 levels in media from cells exposed to WT mitochondria were not significantly different from that of control (media alone) (Fig. 6a). On the

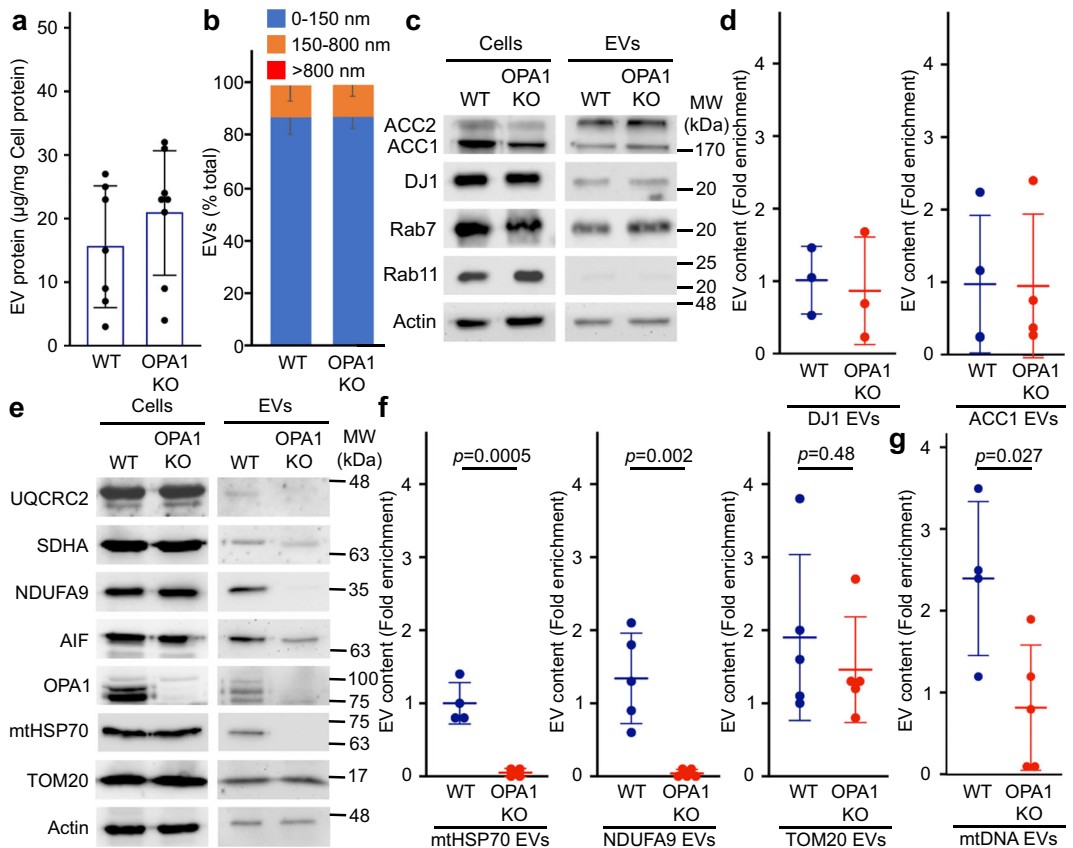

**Fig. 5 OPA1 regulates the inclusion of IM/matrix proteins and mtDNA into EVs. a** Quantification of EV protein yields relative to cellular protein content in WT and OPA1 KO EVs isolated as in Fig. 1. Each point represents one experiment ($n = 7$). Bars show the average ± SD. **b** OPA1 deletion does not alter overall EV size. WT and OPA1 KO EV size was analyzed by EM and binned as indicated. The average percent of EVs in each category ± SD is shown ($n = 3$). No EVs larger than 800 nm were detected. **c**, **d** The inclusion of cytoplasmic proteins into EVs is not altered following OPA1 deletion. Representative western blot (**c**) and quantification (**d**) as in Fig. 1. Individual points represent independent experiments ($n = 3$). Bars show the average ± SD. **e–g** The inclusion of IM/matrix proteins, but not TOM20, into EVs is prevented by OPA1 deletion. Representative western (**e**) and quantification as in Fig. 1 (**f**). mtDNA was quantified by PCR (**g**) Individual points represent independent experiments. Bars show the average ± SD. Two-sided *t*-test. WT (blue), OPA1 KO (red).

other hand, OPA1 KO mitochondria caused a significant increase in IP10 secretion (Fig. 6a), consistent with OPA KO mitochondria being oxidised. This contrasts with the response to EVs, where OPA1 KO EVs did not further increase IP10 secretion (Fig. 6b). In fact, IP10 levels were lower with KO EVs in each of our experiments (Fig. 6b), suggesting that KO EVs are actually less pro-inflammatory than their WT counterpart. This is also supported by the observation that the mRNA expression of the two IFN-dependent genes *Rsad2* and *mIfit1* was significantly lower in cells exposed to OPA1 KO EVs compared to those exposed to WT EVs (Fig. 6c). In contrast, the strong IL6 secretion caused by WT mitochondria was not further stimulated by OPA1 KO mitochondria while neither EV types caused significant IL6 secretion (Fig. 6d, e), suggesting that the response to oxidized mitochondrial content is IP10-selective. Altogether, these results indicate that inhibiting the release of oxidised mitochondrial proteins is sufficient to reduce the capacity of EV to stimulate IP-0 production.

**Oxidative damage to mitochondria blocks the inclusion of mitochondrial proteins within EVs.** We next determined how oxidative damage to mitochondria inhibits the constitutive, MDV-dependent secretion of mitochondrial proteins in EVs. Different classes of MDV have been identified based on cargo selectivity and destination. These include Snx9-dependent MDV

that deliver mitochondrial antigens to endosome for antigen presentation[29], and ROS-induced MDVs that deliver oxidized mitochondrial cargo to lysosomes for degradation[36]. We thus determined whether activation of the ROS-induced pathway targeted mitochondrial content to lysosomes at the expense of the EV pathway (Fig. 3a) in MEFs. Treatment of WT MEFs with AA induced the formation of mtHSP70 and TOM20 MDVs (Fig. 7a) and, consistent with MDVs containing oxidised proteins being delivered to lysosomes for degradation, AA treatment also increased the proportion of mtHSP70 MDVs associating with the lysosomal marker LAMP1 (in the presence of lysosomal inhibitors E64 and Pepstatin A to prevent cargo degradation; Fig. 7b, Supplementary Fig. 4A). Similarly, while the total number of mtHSP70 MDVs was lower in OPA1 KO cells (Fig. 4d), they were positive for LAMP1 to a larger extend than WT mtHSP70 MDVs (Fig. 7c), indicating that the few IM/matrix MDVs remaining in OPA1 KO MEFs are preferentially targeted to lysosomes for degradation. Overall, as mtHSP70 inclusion into EVs is blocked under these conditions (Figs. 1f, 5f), our results suggest that the EV-targeted MDV pool is inhibited by the activation of the damage-induced pathway.

We thus next determined whether directly stimulating the lysosomal pathway in the absence of mitochondrial damage is sufficient to inhibit the inclusion of IM/matrix proteins into EVs. For this, we took advantage of the Parkinson's Disease-related

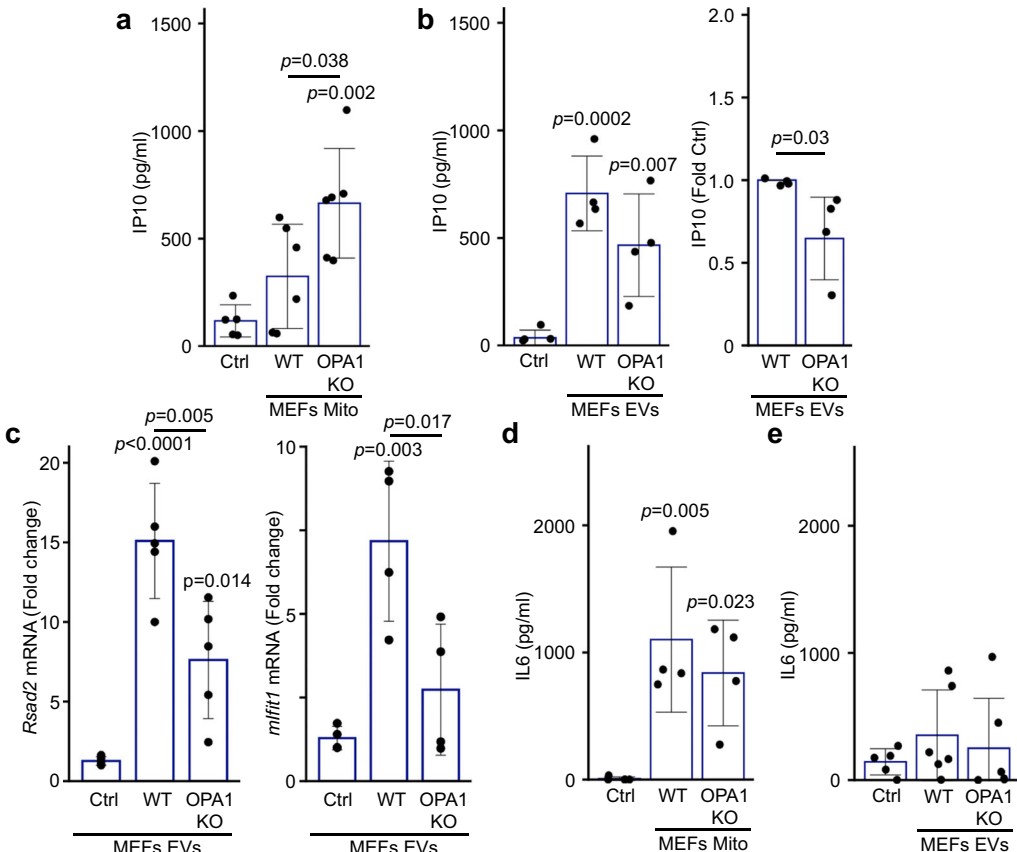

**Fig. 6 OPA1 KO mitochondria but not EVs isolated from the same cells induce an IP10 inflammatory response. a, b** RAW cells were treated as in Fig. 1 with mitochondria (Mito, 12 μg) (**a**) or EVs (from 10 × 10^6 cells) (**b**) isolated from WT or OPA1 KO MEFs and IP10 release in culture media measured by ELISA. Individual points represent actual concentrations (or normalised to WT for (**b**), right panel) in independent experiments (Mito, n = 6; EVs, n = 4). Bars show the average ± SD. One-way ANOVA, two-sided t-test for (**b**), right panel. **c** OPA1 KO EVs show reduced expression of IFN-dependent genes. RAW cells were treated as in (**a–b**) and mRNA levels of the IFN-dependent genes *Rsad2* (left) and *mlfit1* (right) measured by qPCR. Individual points represent independent experiments (*Rsad2*, n = 6; *mlfit1*, n = 4). Bars show the average ± SD. One-way ANOVA. **d, e** RAW cells were treated as in (**a, b**) and IL6 was measured in the culture media by ELISA. Individual points represent independent experiments (Mito, n = 4, EVs, n = 5). Bars show the average ± SD. One-way ANOVA.

protein Parkin, which stimulates the formation of lysosome-targeted MDVs and, importantly, inhibits the Snx9-dependent pathway in the context of antigen presentation[29]. As the cell lines used in our experiments express very low or undetectable levels of Parkin (Supplementary Fig. 4B), we used U2OS cells stably expressing GFP-Parkin to drive the formation of lysosome-directed MDVs, as previously done by us and others to study Parkin-dependent MDVs[30,45]. Consistent with its role in the formation of MDVs destined for lysosomes[36], increased Parkin expression stimulated MDV formation under both basal and AA-stimulated conditions (Fig. 7d) but did not alter the intracellular levels of mitochondrial proteins (Supplementary Fig. 4C). In addition, GFP-Parkin partially associated with mtHSP70 MDVs in AA-treated U2OS cells stably expressing GFP-Parkin (Fig. 7e, Supplementary Fig. 4D), similar to other matrix cargos[45]. We then measured the inclusion of mitochondrial proteins into EVs. Consistent with Parkin stimulating the delivery of mitochondrial proteins to lysosomes at the expense of EVs, mtHSP70 and NDUFA9 were significantly decreased in EVs from GFP-Parkin compared to GFP-expressing cells (Fig. 7f, Supplementary Fig. 4C). However, TOM20 inclusion into EVs was not significantly affected (Fig. 7f, Supplementary Fig. 4C). Altogether, these results indicate that Parkin inhibits the recruitment of mitochondrial proteins to EVs by promoting instead their lysosomal degradation.

## Discussion

Cells secrete mitochondrial proteins and mtDNA into their environment. This has been proposed to serve many functions, including serving as a form of quality control, participating in long-range metabolic regulation or stimulating the immune system[10,18,21,25,27,34,46–48]. However, the underlying mechanisms have remained for the most part elusive. Here we show that MDVs are required for the release of mitochondrial proteins in EVs. This is blocked upon mitochondrial damage, preventing the release of pro-inflammatory oxidised mitochondrial content.

A number of mitochondria-secreted factors behave as DAMPs that are associated with inflammatory cell activation during various pathological conditions. In these conditions, mtDNA and other mitochondrial components (cardiolipin, formyl peptides) can be released in extracellular media and act as an activator for immune cells to stimulate clearance of damaged cells and activate proinflammatory responses[49,50]. Importantly, these studies used LPS stimulation to trigger the release of mitochondrial DAMPs. For example, Joshi et al showed that damaged (low ATP, low membrane potential) extracellular mitochondria released from LPS-activated microglia acts as effectors of the innate immune system by targeting adjacent neurons and astrocytes[47]. Similarly, a number of studies have demonstrated that extracellular DAMPs act as a feedback loop for neutrophil activation[17,51,52]. On the other hand, several studies including this report have shown that

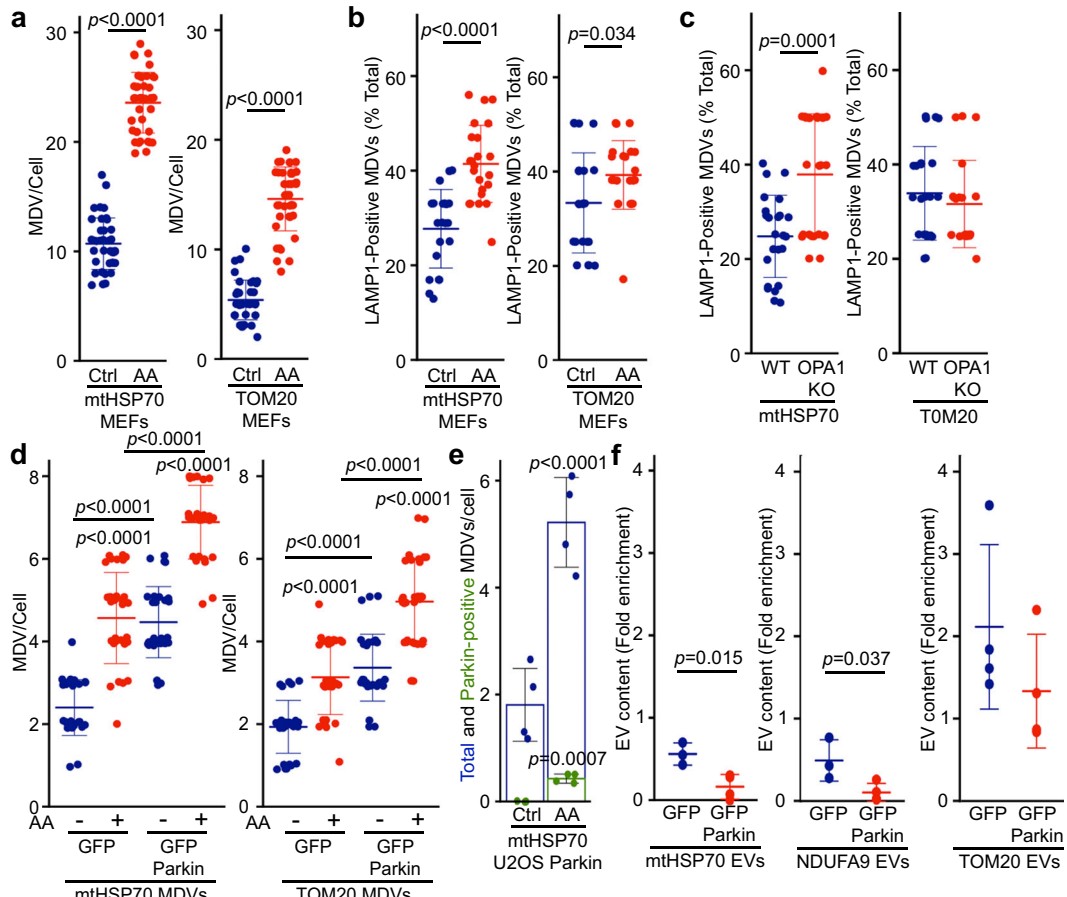

**Fig. 7 Parkin activation inhibits the inclusion of mitochondrial proteins within EVs. a** AA treatment induces MDV formation. WT MEFs grown on coverslips were incubated in the absence (blue) or presence of AA for 1 h (red). mtHSP70 and TOM20 MDVs were then quantified. Each data point represents one cell. Bars represent the average of at least 30 cells in a minimum of three independent experiments ± SD. Two-sided *t*-test. **b**, **c** Damage-induced MDVs are targeted to lysosomes. **b** WT MEFs were treated as in (**a**) in the presence of E64/Pepstatin and MDV associaton with the lysosomal marker LAMP1 quantified as the fraction of total MDVs associated with LAMP1. Alternatively (**c**) LAMP1-positive MDVs were quantified from OPA1 KO MEFs. Each data point represents one cell. Bars represent the average of at least 20 cells in a minimum of three independent experiments ± SD. Two-sided *t*-test. **d** Stable expression of GFP-Parkin increase MDV formation. U2OS cells stably expressing GFP (blue) or GFP-Parkin (red) were treated with AA for 4 h and MDVs quantified. Each data point represents one cell. Bars represent the average of at least 20 cells in a minimum of three independent experiments ± SD one-way ANOVA. **e** mtHSP70 MDVs association with Parkin. U2OS cells stably transfected with GFP-Parkin were treated as in (**d**) and the association of mtHSP70 MDVs quantified by immunofluorescence. Individual points represent independent experiments (*n* = 4). Bars show the average percent of the total (blue) and Parkin-positive (green) mtHSP70 MDVs ± SD. Two-sided *t*-test. **f** GFP-Parkin expression impairs the inclusion of IM/matrix proteins into EVs. EVs were isolated from U2OS cells stably expressing GFP (blue) or GFP-Parkin (red) and the indicated mitochondrial proteins measured as in Fig. 1. Individual points represent independent experiments (*n* = 3, except for TOM20 where *n* = 4). Bars show the average ±SD. Two-sided *t*-test.

mitochondrial proteins can be secreted in EVs in the absence of pro-inflammatory signals[9,28], but the role or mechanism of release remains poorly understood.

To study this question, we first quantified a number of mitochondrial proteins present in EVs. Consistent with a previous report showing selective enrichment of mitochondrial proteins in cancer cell lines[9], our results demonstrated that mitochondrial proteins are selectively incorporated into EVs under non-stimulated conditions. This process is dependent on MDVs, small cargo-selective vesicles that target mitochondrial proteins to other intracellular locations, including lysosomes where they are degraded[36,37]. Specifically, inhibition of IM/matrix MDV formation by either knocking down Snx9 or deleting OPA1 inhibited the inclusion of these proteins in EVs.

While MDV cargo is selected based on target destination and the nature of mitochondrial stress, the mechanisms defining MDV mitochondrial cargo selection are still unclear. Our results

show that the same protein cargo (mtHSP70) can be targeted to EVs or lysosomal degradation depending on the stimulus. Specifically, we observed that upon mitochondrial damage, a greater number of MDVs get transported to lysosome which in turn blocks the secretion of mitochondrial proteins within EVs. The formation of MDVs containing damaged oxidized proteins is stimulated by the PINK/Parkin pathway[45]. Consistent with this, Parkin expression inhibited the secretion of IM/matrix protein in EVs, further demonstrating that ROS-induced MDVs prevent the secretion of oxidised mitochondrial proteins by targeting them to lysosomes for degradation.

This dual-targeting mechanism requiring Snx9 but blocked by Parkin is similar to the mechanism by which professional antigen-presenting cells present mitochondrial antigens (mitAP)[29]. In these studies, stimulation of immune cells (heat shock, LPS or bacteria) stimulates Snx9-dependent mitAP. This was exacerbated by the loss of the PINK1/Parkin pathway, leading to increased

inflammation and Parkinsonian symptoms[53–55]. Here, we found that in cells that are not professional antigen-presenting cells (fibroblasts, epithelial cells), Snx9 is required for the basal secretion of mitochondrial proteins in EVs, and that this is not associated with the strong IL6 response elicited by extracellular mitochondria. Rather, EVs selectively activate an IP10 response that is decreased following mitochondrial oxidative stress. In this context, the role of the PINK1/Parkin pathway would thus be to prevent the release and pro-inflammatory role of oxidised mitochondrial components. This could be particularly important given that oxidised mitochondrial components are more immunogenic and trigger a larger IP10 response (AA, Fig. 1d; OPA1 KO, Fig. 6a, b)[21,56,57] and suggests that, as with bacterial infection[58], mutations in the PINK1/Parkin pathway could lead to increased inflammation following oxidative damage to mitochondria. Nevertheless, other ROS-induced, lysosome-targeted MDV pathways must exist as these MDVs can be induced in a number of cell lines with undetectable Parkin levels (MEFs (Fig. 7a), U2OS (Fig. 7d), HeLa cells[36].

Altogether, our results demonstrate that cells constitutively incorporate mitochondrial DNA and proteins in their EVs in the absence of an IL6-dependent inflammatory response. One remaining question concerns their physiological role. Previous work suggested that secretion of mitochondrial proteins could be a form of quality control where cells export their damaged mitochondria destined for degradation in distant cells[7,47,48]. However, given that lysosome-targeted MDVs prevent the release of oxidised mitochondrial components, this is unlikely to be a widespread phenomenon. On the other hand, an increased number of studies have demonstrated the transfer of mitochondrial content (including mtDNA) between cells, affecting the metabolic output of the recipient cells and promoting tumour growth[59–61]. In this context, the mitochondrial quality control mechanism identified here would serve the dual role of preventing inflammation and ensuring that only functional mitochondrial components are transferred.

## Methods

**Cell culture and treatments**. The following cell lines were used: WT and OPA1 KO MEFs (gift from Dr. Luca Scorrano, University of Padua) including a line with OPA1 reintroduced that we previously described[38], U2OS cells stably transfected with pcDNA3-GFP and GFP-Parkin (gift from Edward Fon) and the mouse macrophage cell line Raw 264.7 (ATCC). All cells were cultured in Dulbecco's modified Eagle's medium (DMEM) (Wisent) supplemented with 10% fetal bovine serum. Where indicated, cells were treated with Antimycin A (AA) (50 μM) for 24 h and samples were collected for Immunofluorescence and western blots. For lysosomal inhibition experiments, cells were treated with combination of E64 (10 μM) and Pepstatin-A (20 μM) with or without AA for 1 h and processed for immunofluorescence. Cell viability was measured by trypan blue exclusion. For suppression of endogenous Snx9 levels in WT MEFs, Snx9 siRNA (Silencer™ Select IDs s83576 and s83577) and negative control siRNA (Silencer™ Select Negative Control No. 1 and 2 siRNA) were used (Thermo life technology). Cells were allowed to adhere on plate/coverslips for 24 h and then transfected with 20 nm of siRNA using siLentFect™ lipid reagent (Bio-Rad) for 24 h.

**EV collection and mitochondria isolation**. Twenty-four hours after seeding cells in 150 mm plates, media was replaced with DMEM supplemented with FBS that had been heat-inactivated and ultracentrifuged at $100,000 \times g$ speed for 9 h to remove EV content. Media (15 ml) and cells ($20 \times 10^6$) were collected 24 h later for EV isolation. To isolate EVs, media was centrifuged at $400 \times g$ for 5 min to remove dead cells. EVs were then pelleted by ultracentrifugation at $100,000 \times g$ for 1 h (Rotor 70.1Ti). The pellet was washed once with PBS. For western blot analysis, EVs were stored at −20 °C before use. For ELISAs and mtDNA quantification, EVs were used immediately. For trypsin digestion, 5 μg of EVs were digested with trypsin (0.01 μg/μl) in HIM buffer (200 mM mannitol, 70 mM sucrose, 10 mM HEPES, pH 7.4, 1 mM EGTA) for 20 min on ice. The reaction was then stopped with soybean trypsin inhibitor.

Mitochondria were isolated as previously described[62]. Briefly, cells were harvested, resuspended in 200 mM mannitol, 70 mM sucrose, 10 mM HEPES, pH 7.5, 1 mM EGTA then broken by passing them 15 times through a 25-gauge needle. Nuclei and cell debris were removed by centrifugation at $1000 \times g$ and the

mitochondrial fraction isolated by centrifugation at $9000 \times g$. The mitochondrial pellet was washed once with the same buffer before use.

**Electron microscopy**. EVs collected from ultracentrifugation were fixed in 2% paraformaldehyde at room temperature. The pellets were then shipped in fixative to Mount Sinai Hospital (Toronto) for processing. There, samples were first incubated in 1% glutaraldehyde for 5 min, then contrasted in a solution of uranyl oxalate (pH 7) before contrasting and embedding in a mixture of 4% uranyl acetate and 2% methyl cellulose in a ratio of 1:9. Images were acquired using EMS 208S (Philips) by an operator that was blinded to the experiment (4–5 fields per grid for a total of 50–60 EVs/grid). EVs size analysis (diameter at widest point) was performed using ImageJ software.

**Antibodies and immunoblots**. The following antibodies were used: mouse anti-actin (A-5316, Sigma-Aldrich); mouse anti-Cytochrome C (SC-13156), rat anti-LAMP1 (SC-19992), goat anti-AIF (SC-9416), rabbit anti-TOM20 (SC-11415), mouse anti-SDHA (SC-390381) and mouse anti-CD9 (SC-13118) (Santa Cruz Biotechnology, Inc.); mouse anti-NDUFA9 (ab14713), rabbit anti-DJ1 (ab18257), mouse anti PDH1 (ab110330), mouse anti-UQCRC2 (ab14745), mouse anti-Parkin (ab77924), rabbit anti-tubulin (ab52866), rabbit anti-Lamin B1 (ab133741) and mouse anti-Alix (ab117600) (Abcam); rabbit anti-ACC (3676-P), rabbit anti-Rab7 (9367-S), and rabbit anti-Rab11 (5589-P) (Cell Signalling Technologies); mouse anti-mtHSP70 (MA3-028) (Thermo scientific); mouse anti-OPA1 (612607) (BD Biosciences); rabbit anti-SNX9 (15721-1-AP) (proteintech). Secondary antibodies were from Jackson Immunoresearch (Alexa Fluor® 488 Donkey Anti-Mouse (715-545-150), Alexa Fluor® Cy-3 Donkey Anti-Mouse (715-165-150), Alexa Fluor® Cy-3 Donkey Anti-Rabbit (711-165-152), Alexa Fluor® 647 Donkey Anti-Rabbit (711-605-152), Alexa Fluor® 647 Donkey Anti-Rat (712-605-153), HRP Goat Anti-Rabbit (111-035-003), HRP Donkey Anti-Goat (705-035-003)), except HRP Anti-mouse that was from Cell Signalling Technology (7076S).

EVs and whole-cell lysates were lysed in 10 mm Tris-HCl, pH 7.9, 150 mm NaCl, 1 mm EDTA, 1% Triton X-100, 50 mm sodium fluoride. Triton-insoluble material was then pelleted at $15,890 \times g$ for 10 min and protein concentration measured with the DC assay (Bio-rad). For immunoblot analysis, 5 μg EVs and 20 μg whole-cell lysates were diluted in 1× Laemmli buffer supplemented with ß-mercaptoethanol, then subjected to SDS-PAGE, transferred to nitrocellulose membranes and blotted with the indicated antibodies (1/1000, except actin at 1/10,000). Membranes were then incubated with a 1:5000 dilution of horseradish peroxidase-conjugated secondary antibodies (Jackson Immunoresearch) and visualized by enhanced chemiluminescence (Thermo scientific) using a Bio-Rad imaging system. Samples were quantified using ImageJ (National Institutes of Health). Uncropped blots can be found in the source data file.

**Immunofluorescence**. Cells were seeded overnight onto glass coverslips (Fischer Scientific) in 24-well plates, then treated as indicated and fixed with 4% paraformaldehyde for 10 min at room temperature (RT). Fixed cells were washed with PBS, permeabilized with 0.2% Triton X-100 in PBS and blocked with 1% BSA in 0.1% Triton X-100 in PBS. Coverslips were incubated with primary antibodies (1/200, except for NDUFA9 at 1/100), followed by incubation with fluorescent conjugated secondary antibodies (1/250, Jackson Immunoresearch). Images were acquired with a Leica SP8 laser scanning fluorescence confocal microscope equipped with a ×63 oil immersion objective using accompanying software (LAS AF). Images were quantified using ImageJ. MDVs were manually defined as circular vesicles significantly smaller in size than mitochondria (typically <150 nm vs 400–500 nm diameter for mitochondria) that were one mitochondrial marker was present but a second one completely absent as judged by the absence of fluorescence signal. These MDVs were considered to be associated with the lysosomal marker LAMP1 if they were clearly within a LAMP1-positive vesicle. Similarly, mtHSP70 MDVs were considered to be associated with Parkin if they were clearly within a GFP-Parkin-positive vesicle. Mitochondrial vesicles close to the plasma membrane were defined as for MDVs, except that only vesicles within 1 μm of the plasma membrane (as defined by the edge of the GFP staining) and 1 μm away from the main mitochondrial network were counted.

**Mitochondrial DNA quantification**. Total DNA from cells and EVs were isolated using a silica-based column DNA purification kit (Purelink DNA isolation kit) following the manufacturer's instructions. Relative mitochondrial DNA levels was measured by quantitative PCR using PowerUp™ SYBR™ Green (Applied biosystems™). Independent reactions were performed for the mtDNA gene cytochrome $c$ oxidase 1[24] and the nuclear gene coding for the 18S ribosomal RNA. Primer sequences are provided in Supplemental Table 1. mtDNA levels were calculated using the CFX Manager™ software in a Bio-Rad CFX Real-Time PCR system.

**Quantification of inflammatory responses**. The proinflammatory effects of extracellular mitochondrial components were tested by adding isolated EVs (EVs from $10 \times 10^6$ cells) or whole mitochondria (12 μg, except where indicated) to $0.2 \times 10^6$ RAW264.7 cells for 16 h. Media and cells were then collected. The presence of IP10 in the media was then determined by ELISA using the Mouse CXCL10/IP-10/CRG2 DuoSet ELISA from the R&D system (cat. DY466) according to the

manufacturer's instructions. The presence of IL-6 in the media was then determined by ELISA using the Mouse IL-6 OptEIA™ ELISA set from BD biosciences (cat. 555240) according to the manufacturer's instructions. For mRNA quantification, cellular RNA was extracted using the GENEzol™ TriRNA pure kit (Geneaid, Cat No. GZX100), from which cDNAs were generated using a high-capacity cDNA reverse transcript kit (Applied biosystems, Cat No. LS4368814). qPCRs were done using the Sensifast SYBR No-ROX kit (Bioline Cat no. BIO-98005) for the following genes (primer sequences in Supplementary Table 1): *Rsad2*, *mIfit1*, and actin as a loading control.

**Statistics and reproducibility**. All immunofluorescence data were quantified and images representative of at least three independent experiments shown (exact n are in the quantification figures). For western blots, most experiments are quantified as described in the appropriate figures. For the few experiments that were not quantified (i.e. control experiments showing enrichment of EV markers, mitochondrial fractionation, the efficiency of siRNAs), the images shown are representative of at least three independent experiments. For western blots and mtDNA quantification, EV content was calculated by measuring the ratio of EV content to cellular content normalized to the equivalent amount of total material for each fraction. For experiments where different treatments/cells were compared, the amount of each tested protein/DNA was first normalized in each fraction (cells, EVs) to that of the control (Actin for proteins, 18S for DNA) before calculating the ratios. Individual experiments are shown along with the average and SD. For MDV quantification, quantification is shown for at least 20 individual cells within three experiments, along with the average and SD. Statistical significance was determined using Student's *t* test (between two groups) or one-way ANOVA with a Tukey post hoc test (multiple comparisons). Complete data and statistics for all quantifications can be found in the Source Data file.

**Reporting summary**. Further information on research design is available in the Nature Research Reporting Summary linked to this article.

## Data availability
The authors declare that the data supporting the findings of this study are available within the paper and its supplementary information files. Source data for all western blot images and quantifications are provided in the Source data file. Source data are provided with this paper.

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

## Acknowledgements

We thank Ruth Slack, David Patten, Mireille Khacho, Heidi McBride, Hema Saranya Ilamathi, and Priya Gatti for insightful comments on the manuscript. This work was supported by grants from the Natural Sciences and Engineering Research Council of Canada and the Fondation UQTR. K.T. is a recipient of a Queen Elizabeth II Diamond Jubilee scholarship and a Fonds du Québec-Santé scholarship. L.C. is a recipient of a Fonds du Québec-Santé scholarships.

## Author contributions

Experiments were designed and data analysed by K.T., G.P. and M.G. K.T., L.C., V.D. and F.E.M. performed experiments. K.T. and M.G. wrote the manuscript, which was critically reviewed by all authors.

## Competing interests

The authors declare no competing interests.
