## [Peer Review File · Nature Communications]

REVIEWER COMMENTS

Reviewer #1 (Remarks to the Author):

Todkar et al. investigate the mechanisms by which mitochondrial contents are sorted into extracellular vesicles (EVs) via mitochondria-derived vesicles (MDVs) during homeostasis. The authors claim that mitochondrial contents packaged into EVs under normal conditions are not pro-inflammatory. Further, they demonstrate that specific mitochondrial proteins are sorted into EVs while others, which are still packaged in MDVs, are not. They also show that inner mitochondrial membrane (IMM) fusion regulator OPA1 is important for sorting IMM and mitochondrial matrix proteins into MDVs and EVs. Additionally, they show that Sorting Nexin-9 (Snx9) contributes to the release of inner mitochondrial membrane (IMM)-derived MDVs as EVs. Finally, they provide evidence that the Parkin MDV pathway limits the sorting of specific mitochondrial proteins into EVs.

MDVs and EVs have key intracellular and intercellular signaling roles, respectively. The interplay between these two subcellular compartments, especially at the level of biogenesis and intercellular signaling, is ill-defined. Moreover, the molecular determinants of mitochondrial cargo sorting into different vesicles is a major open question, and these novel findings will notably advance the field. Major strengths of this well written manuscript are the rigorous quantitative methodology and statistical analysis, as well as the genetic approach used to define critical regulators of mitochondrial content release into EVs. Concerns related to data interpretation and rigor and reproducibility of analytical techniques somewhat lessen enthusiasm for this manuscript in its current form.

Major points

1. Authors inconsistently define pro-inflammatory cytokines.

On lines 79-80, the authors state that they are “monitoring two different pro-inflammatory pathways (irf3 using IP10, and NF- κ B using IL6).” However, Fig. 1 and Fig 6 titles claims that mitochondrial content in EVs is not pro-inflammatory, yet the data show that that both EVs and exogenous mitochondria induce IP10/CXCL10, which can have pro-inflammatory effects. IP10 is secreted by immune cells to promote inflammation during infection, cancer, and inflammatory disease (PMID: 21802343). Relevant to this point, Fig. 6 qRT-PCR of two IFN-dependent genes suggests that Type I IFN is likely induced by EVs, somewhat undermining the idea that WT EVs are not pro-inflammatory. An IFN beta ELISA should be performed to test this point directly. Both IP10 and Type I IFNs can have pro- or anti-inflammatory effects, so accordingly the conclusions should be nuanced. A conclusion more consistent with the authors’ data is that EVs do not induce the NF- κ B regulated cytokine IL-6 in the same way as exogenous mitochondria. However, the authors should take care not to overinterpret this conclusion by extending to global pro-inflammatory cytokine responses or even other NF- κ B regulated cytokine responses since specific NF- κ B stimulated genes can be differentially regulated (PMID: 21772277).

2. Validation of critical reagents

In Fig 1, the authors stimulate RAW264.7 cells with exogenous mitochondria or extracellular vesicles (EVs). A thorough characterization of the mitochondrial fraction was not reported, but is needed because isolation protocols and yields can vary considerably between laboratories and experimental systems. A supplemental figure which shows validation of the isolated mitochondrial fraction is needed. For mitochondria, validation may be accomplished with an immunoblot targeting a cytosolic protein and a mitochondrial protein (e.g. pyruvate dehydrogenase or citrate synthase). Second, Antimycin A is added to stimulate mitochondrial ROS. Results from Antimycin A treatment would be more convincing with validation of an increase in mitochondrial reactive oxygen species (ROS) or at least an increase in total cellular ROS in these experimental conditions. In Fig. 7, the authors draw parallels between mitochondrial damage due to AA treatment or OPA KO increasing MDV delivery to lysosomes. Treatment of AA to the WT and OPA1 KO cells in 7D would add more rigor to this experiment, or at minimally show validation of mitochondrial damage in the OPA1 KO cells.

3. Quantitation of mitochondria-derived vesicles is not clear and immunofluorescence images are missing

Explicit criteria for analysis of MDV should be described. The methods section (line 376) only states "Image [sic] were quantified using ImageJ." A more detailed description of this analysis must be included especially because 4/7 figures rely on this assay, and this analysis is often presented instead of images (e.g. Fig 4.) Additionally, the technical method used in Fig 4 is not described in the text or the figure legend. Presumably, this is an immunofluorescence assay similar to that used in Fig 3, which should be explicitly stated with representative images. Representative images are required for two main reasons: 1. No previous IF assays with PDH or NDUFA9 are presented, 2. MDV quantitation is not described (see above). Similarly, representative images are required for Fig 7 (either in Fig. 7 or in the supplement). The quantitative method for distinguishing LAMP1-positive MDV should be reported as part of the expanded image analysis section of the methods. These protocol details are critical for reproducibility. Lastly, quantitative data should be provided for the Parkin co-localization in Fig. 7B.

Minor points

a. The title of the article uses the phrase "prevent inflammation," which could imply an anti-inflammatory function for MDV. Observations in this manuscript are more consistent with a selective packaging of mitochondrial EVs to avoid release of mtDAMPS in EVs under homeostatic conditions. Authors should consider rephrasing the article title to be more consistent with their observations.

b. I considered it a strength that some observations held true in multiple cell lines/cell types (U2OS, RAW, MEFs) as shown in Fig. 1. However, the authors frequently switched to using one cell line or another throughout the manuscript, without consistently providing a clear rationale. This clarification would be helpful to the reader, as well as labeling the cell type in a given assay directly on graphs (this information is in the legends).

c. At least one representative immunoblot should be shown in the main figures or in the supplement for every experiment which uses immunoblotting (e.g. these are missing in Fig 3). Molecular weights (MW) should be labeled. Labeling MW is especially critical for Fig 5 because the OPA1 KO immunoblot shows part of a band for OPA1.

d. For Fig 3, Change "are required for" in the figure title to "contribute to". According to the analysis, the inner membrane protein NDUFA9 is still released in EVs in the context of Snx9 KD. While an incomplete effect may be the result of partial protein-level depletion of Snx9 (Fig 3B), Snx9 cannot be considered necessary definitively without complete loss of IMM protein sorting into EVs. Similarly, for Fig 5 it is reasonable to state based on the KO data that OPA1 is "required for" sorting IMM proteins into EVs, but the data do not support that OPA1 is required for mtDNA sorting, since some experiments showed mtDNA still present in KO EVs, albeit at a lower level than WT. These conclusions should be softened.

e. Cytokine concentration (pg/ml) is used in Fig 1, and in Fig 6 a fold difference between WT and KO is used. Concentration values need to be reported in Fig 6 because the figure title states that "OPA1 KO mitochondria but not EVs isolated from the same cells induce a pro-inflammatory response." In order to demonstrate that EVs isolated from OPA1 cells do not induce IL-6, EVs must stimulate negligible secretion of IL-6 either in raw value (very low concentration) or compared to unstimulated control (vehicle control).

f. This manuscript and previous reports show that PDH is packaged into MDV (PMID: 27345367). However, here it is shown that PDH MDV are not as frequently secreted as EV of other classes. This protein would be a good control for Fig 2G. If proximity to the plasma membrane correlates with secretion, fewer PDH vesicles should be found at the plasma membrane compared to those that are secreted (e.g. mtHSP70).

Reviewer #2 (Remarks to the Author):

In this manuscript, Todkar and colleagues investigate the mechanisms and function of extracellular vesicles (EVs) released with mitochondrial components. They provide interesting evidence to delineate pathways allowing mitochondrial proteins and DNA to be loaded into vesicles derived from mitochondria under normal and stress conditions. Overall I have to say I thought the evidence presented were strong and it was an interesting paper. I do have a number of suggestions I feel should be addressed:

- EVs did not stimulate IL6 production from RAW264.7 cells, and the authors suggest that this suggest that despite containing mitochondrial protein content "they do not activate the strong NF- κ B response associated with mitochondrial DAMPs."

Could this be a 'dose' issue, rather than a functional difference?

- The authors have done a good job with the characterisation of their EVs. The usual 2000g spin to remove dead cells/debris was not carried out, so it is possible that this is responsible for mitochondrial content. In addition, non-conditioned media control should be included in all experiments where possible, in particular in the electron microscopy.

- In the methods the authors state that "The proinflammatory effects of extracellular mitochondrial components was tested by adding isolated EVs (EVs from 20 x 10⁶ cells/1 x 10⁶ cells) or whole mitochondria (120 μ g/1 x 10⁶ cells) isolated as previously described 62 to RAW264.7 cells for 16 hours". The authors should state how many RAW264.7 cells are being treated, so the reader can make a judgement about the equivalence of the EV transfer (i.e. if it's a 1 to 1 ratio of donor cells to recipient cells, or if not what the ratio is).

The discussion of the pro-inflammatory nature of these EVs could be presented more clearly. It does seem clear that EVs from any condition, (normal, antimycin A treated, or OPA1 knockout cells) do not induce the strong NF- κ B inflammatory response (IL6), that is seen for purified mitochondria. However, the EVs do induce an irf3 inflammatory response (IP10), and OPA1 knockout does alter RSAD2 and mIFIT1 expression in EV recipient cells. More discussion on these latter aspects would clarify what the authors intend the takeaway points from this. In addition, the EVs are applied to receiving cells at an apparently very high ratio. Is the effect seen at a lower, more physiological dose? Is it dose responsive? In addition, what does the 120ug mitochondria represent comparatively to the EVs - is this a high or low dose?

-There could also be more detail in the methodology:

- Electron microscopy: How long were the EVs fixed for? How were the samples stained? What is the size measured – diameter at widest point? How many images were taken? How were image positions selected – randomly or operator selected?
- Western blotting protocol: Concentrations of antibodies, buffers used, reducing conditions. Would also like to see addition of uncropped blot with ladders.
- Unclear as to what this graph in Figure 2G represents. The percentage of cells which have at least one protein identified as being close to the plasma membrane? What is the cut-off for "close"? Do cells have more than one?

- Lines 127-130 – "As shown in Figure 2C, mitochondrial proteins, but not actin, were protected from trypsin digestion in EV preparations, unless the membranes were solubilised with Triton X-100 (Fig. 2C; TX100), indicating that mitochondrial proteins are found inside EVs." Would the authors like to comment on the actin result in light of a recent article in JEV suggesting that actin is found inside EVs? (<https://www.tandfonline.com/doi/full/10.1080/20013078.2020.1757209>) This paper should be cited and discussed.

- The Snx9 knockdown experiment should be repeated with a second siRNA to ensure the phenotypic effect on the cells is not caused by an off-target effect.
- The experiments with OPA1 KO MEFs are compelling. I think it would add strength to the results if the authors test whether re-expressing OPA1 with an expression construct rescues the phenotype
- IL6 and IP10 levels in figure 6 as fold values compared to the wild type, but in figure 1 they're given as absolute values. It would be good to show the absolute values in figure 6, to help the reader compare the results across different figures.
- In many graphs e.g. 3D, 4A-D, the data plotted appears to be all technical replicates across multiple independent experiments. Would it not be statistically more appropriate to plot the mean value from each independent experiment?

Reviewer #3 (Remarks to the Author):

Todkar et al have investigated the interesting and largely unexplored process by which cells secrete mitochondrial content in an inflammatory-silent manner. While the purpose of mitochondrial protein and mtDNA expulsion in the absence of inflammation is unclear, this manuscript argues that mitochondrial derived vesicles (MDVs) deliver mitochondrial content into extracellular vesicles (EVs). Interestingly, the intracellular trafficking protein Snx9 and inner membrane fusion protein OPA1 are required for the incorporation of some mitochondrial proteins into EVs. In agreement with previous work, the authors show that Snx9 regulates certain MDVs and now show that OPA1 is also required for the formation of MDVs containing IM/matrix proteins. They also support a role for Parkin in MDV regulation and argue that the parkin pathway actively diverts damaged mitochondrial content away from EVs and into lysosomes, preventing an inflammatory response.

This is an interesting topic and I am struck by the apparent regulation of mitochondrial protein content in EVs. However, I do have some conceptual and methodology concerns that should be addressed to support the claims of this manuscript.

1. The regulation of inflammatory responses by released mitochondrial content is a relevant issue but I am rather confused by the data presented here. When added to cells, EVs do not stimulate IL6 production but they do induce IP10 production to the same degree as isolated mitochondria, leading the authors to state that EVs "can somewhat stimulate inflammation". These data appear at odds with one of the central messages being that mitochondrial proteins in constitutively formed EVs are not pro-inflammatory. The authors should more accurately define inflammatory responses to EVs in the text.
2. For these experiments (Fig. 1A, B, G), do the authors know that the protein content of the added EVs and isolated mitos was similar (EVs from 20x10⁶ cells vs 120 µg mitochondria)? This appears to be crucial to allow comparison of their inflammatory effects. It would also be helpful to present a blot of the mitochondrial fractionation to assess the purity of the mitochondrial fraction.
3. The authors describe "IM/Matrix-derived MDVs", the formation of which they argue is OPA1 dependent. Are they proposing these MDVs lack the OM? How would this be regulated? The authors could clarify this in the text.
4. It is interesting that in line with PINK1/Parkin repression of mitochondrial antigen presentation (Matheoud et al., 2016), Parkin over expression appears to block mtHSP70 and NDUFA9 incorporation into EVs (Fig. 7G- the argument that it does not affect Tom20 incorporation is not convincing however). The authors argue that Parkin directs damaged mitochondrial content away from EV

pathway and into lysosomes upon damage. I assume Parkin is expressed in these cells and prevents the formation of EVs causing inflammation? To substantiate this, the authors should downregulate Parkin and measure EV mitochondrial content in antimycin treated cells. Also, the evidence of Parkin localisation to mtHSP70 MDVs is not convincing (Fig. 7B). The transient nature of localisation cannot be determined from a single image.

5. Immunoblotted mitochondrial protein content in EVs is quantified throughout the manuscript after normalisation to cellular content. Example blots are frequently missing (e.g Fig 3E, Fig 2B) and should include cells and EVs ran on the same gel and membrane (not as in Fig. 5 C and E). Fig. 2B contains the broadest assessment of EV content whereas the rest of the paper focuses on one or two proteins. What happens to the EV marker Alix upon antimycin A treatment, i.e how specific is the change in EV content in response to mitochondrial damage? Comparison of proteomes of EV fractions isolated from cells in the presence or absence of antimycin could directly demonstrate that different EVs are formed, but this might be beyond the scope of this manuscript.

6. It is not always clear what the authors refer to as mitochondrial damage. Is mtHSP70 content restored in EVs upon treatment of AA-treated cells with ROS scavengers?

7. Microscopy images are also missing and should be included alongside quantification (e.g Fig. 4).

8. Raw values in Fig.6 should be presented as in Fig. 1 for clarity. Why does IP10 (Fold WT) value of WT not equal 1?

9. What parameters are used for the quantification of mitochondrial vesicles at the plasma membrane (Fig. 2 G)? Scale bars and arrows are also missing from Fig. 2H.

Response to reviewers:

The reviewers' comments are in italics. All the changes noted in our responses are highlighted in yellow in the manuscript file.

Reviewer #1:

Todkar et al. investigate the mechanisms by which mitochondrial contents are sorted into extracellular vesicles (EVs) via mitochondria-derived vesicles (MDVs) during homeostasis. The authors claim that mitochondrial contents packaged into EVs under normal conditions are not pro-inflammatory. Further, they demonstrate that specific mitochondrial proteins are sorted into EVs while others, which are still packaged in MDVs, are not. They also show that inner mitochondrial membrane (IMM) fusion regulator OPA1 is important for sorting IMM and mitochondrial matrix proteins into MDVs and EVs. Additionally, they show that Sorting Nexin-9 (Snx9) contributes to the release of inner mitochondrial membrane (IMM)-derived MDVs as EVs. Finally, they provide evidence that the Parkin MDV pathway limits the sorting of specific mitochondrial proteins into EVs.

MDVs and EVs have key intracellular and intercellular signaling roles, respectively. The interplay between these two subcellular compartments, especially at the level of biogenesis and intercellular signaling, is ill-defined. Moreover, the molecular determinants of mitochondrial cargo sorting into different vesicles is a major open question, and these novel findings will notably advance the field. Major strengths of this well written manuscript are the rigorous quantitative methodology and statistical analysis, as well as the genetic approach used to define critical regulators of mitochondrial content release into EVs. Concerns related to data interpretation and rigor and reproducibility of analytical techniques somewhat lessen enthusiasm for this manuscript in its current form.

1. Authors inconsistently define pro-inflammatory cytokines.

On lines 79-80, the authors state that they are “monitoring two different pro-inflammatory pathways (irf3 using IP10, and NF-κB using IL6).” However, Fig. 1 and Fig 6 titles claims that mitochondrial content in EVs is not pro-inflammatory, yet the data show that that both EVs and exogenous mitochondria induce IP10/CXCL10, which can have pro-inflammatory effects. IP10 is secreted by immune cells to promote inflammation during infection, cancer, and inflammatory disease (PMID: 21802343). Relevant to this point, Fig. 6 qRT-PCR of two IFN-dependent genes suggests that Type I IFN is likely induced by EVs, somewhat undermining the idea that WT EVs are not pro-inflammatory. An IFN beta ELISA should be performed to test this point directly.

Both IP10 and Type I IFNs can have pro- or anti-inflammatory effects, so accordingly the conclusions should be nuanced. A conclusion more consistent with the authors' data is that EVs do not induce the NF-kB regulated cytokine IL-6 in the same way as exogenous mitochondria. However, the authors should take care not to overinterpret this conclusion by extending to global pro-inflammatory cytokine responses or even other NF-kB regulated cytokine responses since specific NF-kB stimulated genes can be differentially regulated (PMID: 21772277).

Overall, the message we meant to convey in the original manuscript was that EVs from cells with oxidative damage do not show a greater inflammatory response compared to WT EVs, even though their mitochondria show an enhanced response. This is consistent with

the selective inclusion of mitochondrial proteins into EVs we demonstrate here, but the message was obviously lost. Thus, we thoroughly revised this aspect of the manuscript by 1) clearly referring to IP10 or IL6-dependent responses instead of remaining general, 2) revising the text and figure legends to clearly convey the point that the changes that we observed are relative to WT EV/mitochondria, not in absolute terms and 3) performing new experiments to better define the IP10 and IL6 responses.

Specifically, we performed dose response experiments (**New Fig. 1A-B**) and lowered the amount of mitochondria used for the cytokine experiments to match the EV protein content (12 µg mitochondria corresponds to the amount of proteins present in EVs from 10x10⁶ cells) instead of the amount isolated from the same number of cells (120 µg mitochondria isolated from 10x10⁶ cells). Our new results show that there is a differential activation of IP10 and IL6 with mitochondria and EVs: under the conditions we used, mitochondria induced an IL6 response but no significant increase in IP10 while the reverse was true for EVs (**New Fig. 1A-B**). Nevertheless, the key point we want to make is that the IP10 response is increased in AA-treated and OPA1 KO mitochondria (**New Fig. 1D, New Fig. 6A**) but not in EVs isolated from the same cells (**New Fig. 1D, New Fig. 6B**). Thus, our results indicate that the IP10 response stimulated by oxidative damage to mitochondria is selectively blunted in EVs. This has been made clear throughout the manuscript. We also changed the title of **Fig. 1** and **Fig. 6** to reflect these changes.

We did try to measure IFN-β in our samples but could not detect it by ELISA. This could be down to a timing issue or the fact that IFN-β is notoriously difficult to detect. Nevertheless, as we clarified the IP10 response and show that two other IFN-responsive genes (RSAD2 and mIFIT1) are also induced by EVs, we can reasonably say that EVs induce common Interferon-stimulated genes. For consistency, we referred to this response as an IP10 response throughout the text as this is the main cytokine we measured.

See also **Reviewer 2, points 1 and 4; Reviewer 3, point 1.**

2. Validation of critical reagents

In Fig 1, the authors stimulate RAW264.7 cells with exogenous mitochondria or extracellular vesicles (EVs). A thorough characterization of the mitochondrial fraction was not reported, but is needed because isolation protocols and yields can vary considerably between laboratories and experimental systems. A supplemental figure which shows validation of the isolated mitochondrial fraction is needed. For mitochondria, validation may be accomplished with an immunoblot targeting a cytosolic protein and a mitochondrial protein (e.g. pyruvate dehydrogenase or citrate synthase).

The characterisation of isolated mitochondria is now presented in **New Sup. Fig. 1A**

Second, Antimycin A is added to stimulate mitochondrial ROS. Results from Antimycin A treatment would be more convincing with validation of an increase in mitochondrial reactive oxygen species (ROS) or at least an increase in total cellular ROS in these experimental conditions.

Many labs including ours have demonstrated that Antimycin A increases mitochondrial ROS (see for example **Ref 30-32** of the current manuscript). We have published that AA and OPA1 KO significantly increase mitochondrial ROS in the exact same cells as used here (OPA1 WT/KO) using mitoSOX (Demers-Lamarche et al. (2016) J. Biol. Chem; **Ref**

30 in the current manuscript). We referenced it in the text p. 5 to avoid republishing the exact same experiments.

Nevertheless, to further address the relationship between ROS and mitochondrial EV content, we measured mitochondrial EV content in AA-treated MEFs in the presence of the antioxidant NAC, which rescued the inclusion of mtHSP70 into EVs (Sup. Fig. 1F).

In Fig. 7, the authors draw parallels between mitochondrial damage due to AA treatment or OPA1 KO increasing MDV delivery to lysosomes. Treatment of AA to the WT and OPA1 KO cells in 7D would add more rigor to this experiment, or at minimally show validation of mitochondrial damage in the OPA1 KO cells.

As suggested by the reviewer, we treated WT and OPA1 KO MEFs with AA and measured MDVs and LAMP1-associated MDVs. Perhaps unsurprisingly, OPA1 KO MEFs failed to induce MDV formation in response to AA (figure to the right). This is likely the consequence of these cells already having defective mitochondria (PMID: 25298396, PMID: 24055366) and therefore AA not being able to further inhibit Complex III. As this makes it difficult to draw conclusions relative to MDV formation, we did not include this data in the revised manuscript.

3. *Quantitation of mitochondria-derived vesicles is not clear and immunofluorescence images are missing*

Explicit criteria for analysis of MDV should be described. The methods section (line 376) only states “Image [sic] were quantified using ImageJ.” A more detailed description of this analysis must be included especially because 4/7 figures rely on this assay, and this analysis is often presented instead of images (e.g. Fig 4.) Additionally, the technical method used in Fig 4 is not described in the text or the figure legend. Presumably, this is an immunofluorescence assay similar to that used in Fig 3, which should be explicitly stated with representative images. Representative images are required for two main reasons: 1. No previous IF assays with PDH or NDUFA9 are presented, 2. MDV quantitation is not described (see above). Similarly, representative images are required for Fig 7 (either in Fig. 7 or in the supplement). The quantitative method for distinguishing LAMP1-positive MDV should be reported as part of the expanded image analysis section of the methods. These protocol details are critical for reproducibility. Lastly, quantitative data should be provided for the Parkin co-localization in Fig. 7B.

We added details on how we quantified MDVs and their colocalization with LAMP1 in the Methods section (and corrected the typo, **page 18**), and clarified that the quantification in **Fig. 4** was done from immunofluorescence images. As requested, the quantification of mtHSP70 MDVs co-localising with Parkin is now presented in **New Fig. 7E** (the representative IF image has been changed to make clearer and moved to **New Sup. Fig. 4D**). We also added representative images for the quantifications in Figure 4 (**New Sup. Fig. 3**) and Figure 7 (**New Sup. Fig. 4A**).

4. Minor points:

a. *The title of the article uses the phrase “prevent inflammation,” which could imply an anti-inflammatory function for MDV. Observations in this manuscript are more consistent with a selective packaging of mitochondrial EVs to avoid release of mtDAMPS in EVs under homeostatic conditions. Authors should consider rephrasing the article title to be more consistent with their observations.*

As suggested by the reviewer, we have changed the title of the manuscript to “**Selective packaging of mitochondrial proteins into extracellular vesicles prevents the release of mitochondrial DAMPs**”.

b. *I considered it a strength that some observations held true in multiple cell lines/cell types (U2OS, RAW, MEFs) as shown in Fig. 1. However, the authors frequently switched to using one cell line or another throughout the manuscript, without consistently providing a clear rationale. This clarification would be helpful to the reader, as well as labeling the cell type in a given assay directly on graphs (this information is in the legends).*

As suggested by the reviewer, we have now provided a rationale in the text (**p. 5, 7, 11**) for the different cell lines we used and labeled them in the figures.

c. *At least one representative immunoblot should be shown in the main figures or in the supplement for every experiment which uses immunoblotting (e.g. these are missing in Fig 3). Molecular weights (MW) should be labeled. Labeling MW is especially critical for Fig 5 because the OPA1 KO immunoblot shows part of a band for OPA1.*

We have now included WB data showing both cells and EVs for **Fig. 3B** and the original Fig. 7E (**now in New Sup. Fig. 4C**). In addition, all WB figures now show MW markers and original blots can be found in the **Raw data** file. For the OPA1 blot, the extra band was higher than the expected MW for OPA1 (70-90 kDa) and was the result of reprobing the blot with several antibodies to generate all the data present in the figure. The absence of OPA1 in the KO cells can also be seen in the new WB showing the rescue of OPA1 expression in OPA1 put back cells (**New Sup. Fig. 2D**).

d. *For Fig 3, Change “are required for” in the figure title to “contribute to”. According to the analysis, the inner membrane protein NDUFA9 is still released in EVs in the context of Snx9 KD. While an incomplete effect may be the result of partial protein-level depletion of Snx9 (Fig 3B), Snx9 cannot be considered necessary definitively without complete loss of IMM protein sorting into EVs. Similarly, for Fig 5 it is reasonable to state based on the KO data that OPA1 is “required for” sorting IMM proteins into EVs, but the data do not support that OPA1 is required for mtDNA sorting, since some experiments showed mtDNA still present in KO EVs, albeit at a lower level than WT. These conclusions should be softened.*

We agree with the reviewer that the incomplete loss of Snx9 makes it harder to clearly state that the MDV pathway we identified is absolutely required for the release of mitochondrial proteins/DNA within EVs. We therefore made the change requested by the reviewer for Fig. 3 and used “regulate” rather than “is required” throughout the text.

e. Cytokine concentration (pg/ml) is used in Fig 1, and in Fig 6 a fold difference between WT and KO is used. Concentration values need to be reported in Fig 6 because the figure

title states that “OPA1 KO mitochondria but not EVs isolated from the same cells induce a pro-inflammatory response.” In order to demonstrate that EVs isolated from OPA1 cells do not induce IL-6, EVs must stimulate negligible secretion of IL-6 either in raw value (very low concentration) or compared to unstimulated control (vehicle control).

Raw values for IL6 and IP10 are now presented in all experiments shown in **New Fig. 1A-B and and D**, as well as **New Fig. 6A, B, D and E**. We nevertheless added a fold change graph for the quantification of the IP10 response to WT/KO EVs (**New Fig. 6B**) to highlight the fact that IP10 levels were lower for OPA1 KO EVs than WT EVs in all experiments. We did not observe significant IL6 secretion following incubation of RAW cells with EVs (WT or KO; **Fig. 1A, 6E**).

By stating that KO EVs do not induce a pro-inflammatory response, we meant that these EVs did not cause IL6 or IP10 release beyond the basal response observed in WT EVs. As we agree with the reviewer that WT EVs do induce an IP10 response, we revised our interpretation of the data throughout the manuscript to make this clear. The title of **Fig. 6** was also modified to “OPA1 KO mitochondria but not EVs isolated from the same cells selectively induce an IP10 inflammatory response”. See also **Point 1 of reviewers 1-3** for a more detailed description of the changes we made to the inflammation data and its interpretation.

f. This manuscript and previous reports show that PDH is packaged into MDV (PMID: 27345367). However, here it is shown that PDH MDV are not as frequently secreted as EV of other classes. This protein would be a good control for Fig 2G. If proximity to the plasma membrane correlates with secretion, fewer PDH vesicles should be found at the plasma membrane compared to those that are secreted (e.g. mtHSP70).

As suggested by the reviewer, we counted the number of cells containing PDH-positive vesicles close to the plasma membrane which showed a somewhat lower numbers of vesicles than mtHSP70 (**New Fig. 2G**). The data for both mtHSP70 and PDH nevertheless needs to be interpreted with caution given the small number of vesicles involved.

Reviewer #2:

In this manuscript, Todkar and colleagues investigate the mechanisms and function of extracellular vesicles (EVs) released with mitochondrial components. They provide interesting evidence to delineate pathways allowing mitochondrial proteins and DNA to be loaded into vesicles derived from mitochondria under normal and stress conditions. Overall I have to say I thought the evidence presented were strong and it was an interesting paper. I do have a number of suggestions I feel should be addressed:

1. EVs did not stimulate IL6 production from RAW264.7 cells, and the authors suggest that this suggest that despite containing mitochondrial protein content “they do not activate the strong NF-κB response associated with mitochondrial DAMPs. Could this be a ‘dose’ issue, rather than a functional difference?

We performed dose response experiments as suggested by the reviewer (**New Fig. 1A-B**). The amounts we used in the original manuscript were 120 μg mitochondria (corresponding to the amount of mitochondria isolated from 10x10⁶ cells) and EVs isolated from 10x10⁶ cells (around 12 μg proteins). These were incubated with 0.2x10⁶ RAW cells to measure cytokine release. As these amounts were relatively high (especially for mitochondria), we

performed the dose response experiments using 120 μg mitochondria and EVs isolated from 10×10^6 cells as the largest doses. We found that cytokine release was proportional to the amount of material added to cells for conditions where we could detect cytokine secretion (IP10 with EVs and IL6 with mitochondria (**New Fig. 1A-B**). Thus, for the other experiments (AA, **New Fig. 1D**; OPA1 KO, **New Figure 6**), we used amounts that corresponded to the same quantity of protein (12 μg mitochondria and EVs isolated from 10×10^6 cells).

In our hands, EVs did not trigger an IL6 response irrespective of the conditions we used. We thus don't think that this is a "dose" issue but, as we cannot exclude some IL6 secretion below the detection limit of our assay, we stated that we did not see the strong response found in mitochondria rather than claiming no response at all (**page 5**). Nevertheless, the key point here is not the IL6 response but the fact that the EV-associated IP10 response is modulated by mitochondrial oxidative stress. We completely reworked our inflammation experiments to convey this point and removed the references to a general decrease in pro-inflammatory responses (**see also Reviewer 1, point 1; Reviewer 2, point 4**)

2. The authors have done a good job with the characterisation of their EVs. The usual 2000g spin to remove dead cells/debris was not carried out, so it is possible that this is responsible for mitochondrial content. In addition, non-conditioned media control should be included in all experiments where possible, in particular in the electron microscopy.

We did a 400g spin to remove dead cells/large debris and, consequently, did not find these in our EM analysis (**Fig. 2D-F, New Sup. Fig. 1G**). For EV isolation, we based our procedure on Kowal et al. (2016) PNAS (**ref 28, PMID: 26858453**), where following a 300g spin to remove cells, they centrifuged the supernatants at 2K, 10K and 100K. Using this protocol, they demonstrated that the 2K pellet contains larger EVs, not debris or apoptotic bodies. As mitochondrial proteins could potentially be found in more than one EV population, we choose to pull together all EV types by centrifugating all EVs at 100K. Nevertheless, we did isolate EVs using the 2K spin to directly address the reviewer's question and found that mitochondrial proteins are still present in EVs after the 2K pellet is removed (Figure at the right) However, as all the data in the manuscript was generated using EVs isolated without the 2K spin, we did not include this in the manuscript.

As suggested by the reviewer, we added a non-conditioned media control for western blot and EM (**New Sup. Fig. 1D, G**). As expected, we did not detect any mitochondrial content in non-conditioned media.

3. In the methods the authors state that "The proinflammatory effects of extracellular mitochondrial components was tested by adding isolated EVs (EVs from 20×10^6 cells/ 1×10^6 cells) or whole mitochondria ($120 \mu\text{g}/1 \times 10^6$ cells) isolated as previously described 62 to RAW264.7 cells for 16 hours". The authors should state how many RAW264.7 cells are being treated, so the reader can make a judgement about the equivalence of the EV

transfer (i.e. if it's a 1 to 1 ratio of donor cells to recipient cells, or if not what the ratio is).

The recipient cells consisted of 0.2×10^6 RAW cells. These were exposed to EVs isolated from 10×10^6 cells for AA and OPA KO experiments (a ratio similar to that of previous publications e.g. PMID: 29985392, PMID: 32574561). We also present the response to lower amounts in **Fig. 1A-B** (see reviewer 2, point 1 above). This is now properly stated in the methods (p. 18).

4. The discussion of the pro-inflammatory nature of these EVs could be presented more clearly. It does seem clear that EVs from any condition, (normal, antimycin A treated, or OPA1 knockout cells) do not induce the strong NF- κ B inflammatory response (IL6), that is seen for purified mitochondria. However, the EVs do induce an irf3 inflammatory response (IP10), and OPA1 knockout does alter RSAD2 and mIFIT1 expression in EV recipient cells. More discussion on these latter aspects would clarify what the authors intend the takeaway points from this. In addition, the EVs are applied to receiving cells at an apparently very high ratio. Is the effect seen at a lower, more physiological dose? Is it dose responsive? In addition, what does the 120ug mitochondria represent comparatively to the EVs - is this a high or low dose?

The discussion of the inflammation response was indeed confusing. By saying that there was no pro-inflammatory response with OPA1 KO EVs, we actually meant no increase over WT cells, not that there was objectively no response. We have now fixed this throughout the text.

We also did dose response experiments as suggested by the reviewer (**New Fig. 1A-B**). These experiments (using EV ratios within the range of previously published experiments PMID: 29985392, PMID: 32574561) show that the IP10 response to EVs is dose-dependent, but still observed with lower amounts of EVs.

120 μ g of mitochondria represent approximately the amount of mitochondria isolated from the 10×10^6 cells from which the EVs are isolated, but it is 10 times the amount of protein found in these EVs. We thus modified the experimental setup for the OPA1 KO and AA experiments to use 12 μ g mitochondria instead of 120 μ g, and redid all experiments (both mitochondria and EVs). The results are now in **New Fig. 1D, New Fig. 6A, B, D and E**.

See also Reviewer 1, Point 1, Reviewer 2, Point 1 and Reviewer 3, Point 1.

5. There could also be more detail in the methodology:

a. Electron microscopy: How long were the EVs fixed for? How were the samples stained? What is the size measured – diameter at widest point? How many images were taken? How were image positions selected – randomly or operator selected?

EVs were fixed in 2% PBS then shipped in fixative to be processed. At Mount Sinai Hospital (Toronto, Canada), the samples were first incubated in 1% glutaraldehyde for 5 min, then contrasted in a solution of uranyl oxalate (pH 7) before contrasting and embedding in a mixture of 4% uranyl acetate and 2% methyl cellulose in a ratio of 1:9. When we received back the grids, an operator that was blind to the experiment took 4-5 fields per grid for a total of 50-60 EVs/grid, simply looking for regions of the grid where EVs were present. We then measured the diameters at the widest point. This is now clearly stated in the Methods section.

b. Western blotting protocol: Concentrations of antibodies, buffers used, reducing conditions. Would also like to see addition of uncropped blot with ladders.

The method for western blot and immunofluorescence, including antibody concentrations, have been clarified in the Method section.

The uncropped blots with ladders can be found in the **Raw data** file.

c. Unclear as to what this graph in Figure 2G represents. The percentage of cells which have at least one protein identified as being close to the plasma membrane? What is the cut-off for “close”? Do cells have more than one?

The graph in **Fig. 2G** represents the percentage of cells which have at least one vesicle positive for a mitochondrial marker but not the other close to the plasma membrane. This has now been made more explicit in the y axis label. The cut-off was 1 μm , and we considered only vesicles that were away from the main mitochondrial network ($> 1 \mu\text{m}$) to avoid measuring vesicles just budding off mitochondria. Under these conditions, cells with more than 1 vesicle were extremely rare. This has now been made clear in the legend for **Fig. 2** and in the methods.

6. Lines 127-130 – “As shown in Figure 2C, mitochondrial proteins, but not actin, were protected from trypsin digestion in EV preparations, unless the membranes were solubilised with Triton X-100 (Fig. 2C; TX100), indicating that mitochondrial proteins are found inside EVs.” Would the authors like to comment on the actin result in light of a recent article in JEV suggesting that actin is found inside EVs? (<https://www.tandfonline.com/doi/full/10.1080/20013078.2020.1757209>). This paper should be cited and discussed.

The exact localisation of actin within EVs has been controversial. While some studies suggest that it is found outside EVs or in exomeres (PMID: 32795414), others have suggested that it is found inside vesicular structures (i.e. the article cited by the reviewer). However, the procedure used by Choi *et al.* is different from the one we used (it should eliminate exomeres while enriching exosomes), making it difficult to compare results. Therefore, because of the discrepancies in the literature concerning the intravesicular actin content (likely due to differences in the EV population measured) and the fact that measuring actin is not required for this experiment (the proper control is trypsin+detergent), we removed the statement about actin in the text.

7. The Snx9 knockdown experiment should be repeated with a second siRNA to ensure the phenotypic effect on the cells is not caused by an off-target effect.

As suggested by the reviewer, we now present EV data using two distinct Snx9 siRNAs (**New Fig. 3B and E, New Supp. Fig. 2A-B**). Both siRNAs caused a similar reduction in mtHSP70 and NDUFA9 within EVs.

8. The experiments with OPA1 KO MEFs are compelling. I think it would add strength to the results if the authors test whether re-expressing OPA1 with an expression construct rescues the phenotype

We did the experiment as suggested by the reviewer and found that re-expressing OPA1 does rescue the presence of mitochondrial proteins within EVs (**New Supp. Fig. 2D-F**)

9. *IL6 and IP10 levels in figure 6 as fold values compared to the wild type, but in figure 1 they're given as absolute values. It would be good to show the absolute values in figure 6, to help the reader compare the results across different figures.*

Raw values for IL6 and IP10 are now presented in all figures. We however also show the fold change for the OPA1 KO EV experiments as it shows that IP10 secretion was lower for KO EVs in all experiments (**New Fig. 6B**). **See also Reviewer 1 point 4e.**

10. *In many graphs e.g. 3D, 4A-D, the data plotted appears to be all technical replicates across multiple independent experiments. Would it not be statistically more appropriate to plot the mean value from each independent experiment?*

There can be relatively a large variation in the number of MDVs produced in individual cells within each experiment. Thus, to better represent this variability, it has become the standard to plot MDVs for individual cells rather than averaging experiments (see PMID: 32311122, PMID: 27458136, PMID: 27345367). That way, we see the whole population rather than averages.

Reviewer #3

Todkar et al have investigated the interesting and largely unexplored process by which cells secrete mitochondrial content in an inflammatory-silent manner. While the purpose of mitochondrial protein and mtDNA expulsion in the absence of inflammation is unclear, this manuscript argues that mitochondrial derived vesicles (MDVs) deliver mitochondrial content into extracellular vesicles (EVs). Interestingly, the intracellular trafficking protein Snx9 and inner membrane fusion protein OPA1 are required for the incorporation of some mitochondrial proteins into EVs. In agreement with previous work, the authors show that Snx9 regulates certain MDVs and now show that OPA1 is also required for the formation of MDVs containing IM/matrix proteins. They also support a role for Parkin in MDV regulation and argue that the parkin pathway actively diverts damaged mitochondrial content away from EVs and into lysosomes, preventing an inflammatory response.

This is an interesting topic and I am struck by the apparent regulation of mitochondrial protein content in EVs. However, I do have some conceptual and methodology concerns that should be addressed to support the claims of this manuscript.

1. *The regulation of inflammatory responses by released mitochondrial content is a relevant issue but I am rather confused by the data presented here. When added to cells, EVs do not stimulate IL6 production but they do induce IP10 production to the same degree as isolated mitochondria, leading the authors to state that EVs “can somewhat stimulate inflammation”. These data appear at odds with one of the central messages being that mitochondrial proteins in constitutively formed EVs are not pro-inflammatory. The authors should more accurately define inflammatory responses to EVs in the text.*

By saying that there was no pro-inflammatory response with OPA1 KO EVs, we actually meant no increase over WT cells, not that there was objectively no response. As our wording was clearly misleading, we completely revised the text to clearly state that OPA1 KO EVs stimulate a weaker IP10 response than WT EVs, while KO mitochondria cause a greater release of IP10. We have also completely redone the inflammation experiments to

have more comparable amounts of EVs and mitochondria (**New Fig. 1A-B, D; New Fig. 6**). See also **Reviewer 1, point 1; Reviewer 2, points 1 and 4**.

2. For these experiments (Fig. 1A, B, G), do the authors know that the protein content of the added EVs and isolated mitos was similar (EVs from 20x10⁶ cells vs 120 µg mitochondria)? This appears to be crucial to allow comparison of their inflammatory effects. It would also be helpful to present a blot of the mitochondrial fractionation to assess the purity of the mitochondrial fraction.

In fact, we had used different protein amounts because we originally based our comparison on the material obtained from the same number of cells. We now realise that it is better to compare equivalent amount of proteins instead. Thus, we did 1) a dose response experiment to test different amounts of EVs and mitochondria (**New Fig. 1A-B**) and 2) we used EVs from 10 x 10⁶ cells and 12 µg mitochondria (1/10 of the original amounts, corresponding to the amount of proteins found in EVs isolated from 10 x 10⁶ cells) for the AA and OPA1 KO experiments (**New Fig. 1D and New Fig. 6**). Our new results are thus much more comparable. Nevertheless, in the revised version of the manuscript, we refrained as much as possible to make direct comparisons between mitochondria and EVs in terms of absolute responses to focus on the key point: oxidized mitochondria stimulate IP10 secretion to a greater level than Control mitochondria, but EVs from cells with oxidative damage don't. The oxidized material that trigger the IP10 response is thus not transferred to the EVs. **See also Reviewer 1, point 1; Reviewer 2, points 1, 3 and 4.**

We added a blot of the mitochondrial fraction in **New Sup. Fig. 1A**.

3. The authors describe "IM/Matrix-derived MDVs", the formation of which they argue is OPA1 dependent. Are they proposing these MDVs lack the OM? How would this be regulated? The authors could clarify this in the text.

The IM/Matrix MDVs we measured are consistent with those previously described by the McBride lab (reviewed in Sugiura et al.), which contain two membranes (IM and OM) but lack the OM marker TOM20. We clarified the **text on page 8**.

4. It is interesting that in line with PINK1/Parkin repression of mitochondrial antigen presentation (Matheoud et al., 2016), Parkin over expression appears to block mtHSP70 and NDUFA9 incorporation into EVs (Fig. 7G- the argument that it does not affect Tom20 incorporation is not convincing however). The authors argue that Parkin directs damaged mitochondrial content away from EV pathway and into lysosomes upon damage. I assume Parkin is expressed in these cells and prevents the formation of EVs causing inflammation? To substantiate this, the authors should downregulate Parkin and measure EV mitochondrial content in antimycin treated cells. Also, the evidence of Parkin localisation to mtHSP70 MDVs is not convincing (Fig. 7B). The transient nature of localisation cannot be determined from a single image.

We used Parkin mainly as a tool to activate the lysosome-targeted MDV pathway as published by the McBride lab. Nevertheless, to address the reviewer's question, we first measured Parkin expression in the cell lines we used in this manuscript relative to a brain sample, which is known to express high levels of Parkin. The result (**New Sup. Fig. 4B**) showed that these cell lines express very low or undetectable levels of Parkin, making the knockdown experiment very challenging to do.

Despite the undetectable levels of Parkin in MEFs, we observed a robust increase in lysosome-targeted MDVs following AA treatment in these cells (**Fig. 7A-B**). We have now performed similar experiments in U2OS-GFP cells and also found that AA stimulated MDV formation in the absence of GFP-Parkin (**New Fig. 7D**). This suggests that Parkin-independent pathway(s) can target oxidized mitochondrial cargo to lysosomes. This is also consistent with the original MDV report (Soubanier et al, 2012) that showed ROS-induced MDVs in Parkin-null HeLa cells.

Irrespective of the actual role of endogenous Parkin, its overexpression stimulates the formation of lysosome-targeted MDVs and inhibits mitochondrial antigen presentation (Matheoud et al.). We thus used the same overexpression strategy to push the system towards the lysosomal pathway in the absence of actual oxidative damage. Overall, our results with both AA treatment and GFP-Parkin expression support the idea that lysosomal targeting of oxidized MDVs prevents the release of mitochondrial proteins within EVs.

Given the above, we made the following changes to address the reviewer's comment: we performed experiments to demonstrate that U2OS induce MDV formation following AA treatment and that this is increased by GFP-Parkin expression (**New Fig. 7D**), we provided data indicating that the cells we used in this study express very low or undetectable levels of endogenous Parkin (**Sup. Fig. 4B**); we rewrote the last section of the result section and part of the discussion to make it clear that we are not claiming that EV mitochondrial content is completely dependent on Parkin.

Concerning the colocalization of Parkin and mtHSP70, the reviewer is right that we cannot assess the transient nature of Parkin recruitment to mtHSP70. The statement was based on previous published work (McLelland et al. (2014) EMBO J) and the fact that only a small number of mtHSP70 vesicles colocalized with GFP-Parkin. To be clearer, we quantified Parkin-positive mtHSP70 MDVs (**New Fig. 7E**), changed the representative IF image for a clearer one (**New Sup. Fig. 4D**) and removed "transiently" from the sentence in **p. 11**.

5. Immunoblotted mitochondrial protein content in EVs is quantified throughout the manuscript after normalisation to cellular content. Example blots are frequently missing (e.g Fig 3E, Fig 2B) and should include cells and EVs ran on the same gel and membrane (not as in Fig. 5 C and E). Fig. 2B contains the broadest assessment of EV content whereas the rest of the paper focuses on one or two proteins. What happens to the EV marker Alix upon antimycin A treatment, i.e how specific is the change in EV content in response to mitochondrial damage? Comparison of proteomes of EV fractions isolated from cells in the presence or absence of antimycin could directly demonstrate that different EVs are formed, but this might be beyond the scope of this manuscript.

We now show example WB including cells and EVs for all experimental setups. For example, EV data has been added to **New Fig. 3B**. **Fig. 2B** has been quantified from numerous experiments similar to those presented in **Fig. 1C** and **5E**. Although we put an empty space between Cells and EVs in the example WBs presented in the manuscript to distinguish them, all WB were quantified from Cells and EVs ran on the same gel (using the same exposure). This can be seen in the uncropped blots shown in the **Raw Data** file. In the case of **Fig. 5C** and **E**, there was simply a lane between Cells and EVs in the original WB.

As suggested by the reviewer, we also measured Alix EV content in AA-treated MEFs and OPA1 KO MEFs, and did not observe any significant change relative to the control (**New Sup. Fig. 2C**).

6. It is not always clear what the authors refer to as mitochondrial damage. Is mtHSP70 content restored in EVs upon treatment of AA-treated cells with ROS scavengers?

The reviewer is right that this was not clearly explained. By mitochondrial damage, we were actually referring to oxidative stress which is induced with both AA and OPA1 KO (**Refs 30-32** from our manuscript). This has now been clarified throughout the text.

As for the effect of ROS scavengers, NAC did rescue mtHSP70 incorporation into EVs from AA-treated cells (**New Sup. Fig. 1F**), further supporting our hypothesis that oxidative damage triggers the targeting of mitochondrial proteins for mitochondrial degradation, at the expense of the EV pathway.

7. Microscopy images are also missing and should be included alongside quantification (e.g Fig. 4).

Representative images for the quantification in Figure 4 and Figure **7B-C** (colocalization with LAMP1) are now presented in **Sup. Fig 3** and **Sup. Fig. 4A**, respectively.

8. Raw values in Fig.6 should be presented as in Fig. 1 for clarity. Why does IP10 (Fold WT) value of WT not equal 1?

Raw values for IL6 and IP10 are now presented for all experiments (**Fig. 1A, B, D; Fig. 6A, B, D, E**). In addition, we showed the fold change for the IP10 quantification of OPA1 KO EVs to demonstrate that IP10 release was reduced relative to WT in all experiments (**Fig. 6B**).

9. What parameters are used for the quantification of mitochondrial vesicles at the plasma membrane (Fig. 2 G)? Scale bars and arrows are also missing from Fig. 2H.

This has been clarified in the legend for **Fig. 2** and in the methods (see **Reviewer 2 point 5c**). For the scale bar and arrows, they were present in the original TIF figure, but something seems to have happened during the conversion to the pdf file. We made sure that this was fixed for the resubmission.

REVIEWERS' COMMENTS

Reviewer #1 (Remarks to the Author):

Overall, the authors' revisions responded prior concerns well with one point that still needs elucidation to enhance rigor and reproducibility. This can be addressed with minor text revisions.

- Key aspects of MDV analysis from confocal micrographs are still missing. For example, in the previous version of this manuscript, the authors claimed that images were quantified with ImageJ. Now, they claim that some confocal micrographs were assessed qualitatively. In one example, they state that "MDVs were considered to be colocalized with the lysosomal marker LAMP1 if they were clearly within a LAMP1-positive vesicle" This is not a standard colocalization analysis (e.g. Pearson's correlation coefficient or Mander's overlap coefficient) Authors should strongly consider rephrasing colocalization to "association" and clearly stating in the methods that their analysis is qualitative. Additionally, it is not clear how the absence of a mitochondrial marker is defined over background. It should be stated if the fluorescence intensity of these regions was measured or if this was also a qualitative assessment. The study of mitochondria-derived vesicle biology is still a nascent field, and it is critical for reproducibility to be transparent in analytical approaches.

Reviewer #2 (Remarks to the Author):

The authors have done a considerable amount of additional work that has addressed my issues and has significantly improved the manuscript.

Reviewer #3 (Remarks to the Author):

The authors have carefully addressed my concerns and significantly improved the manuscript by providing additional experimental evidence and revising the text. The observation that EVs are largely non-inflammatory is interesting and will guide further experiments to unravel the physiological role of EVs. Moreover, the observation that increased MDV targeting to lysosomes limits the packaging of oxidized mitochondria into EVs is intriguing and sheds new light at the interface of mitochondrial quality control and inflammation, although the sorting mechanism remains unclear. To support their key finding, the authors now compare the inflammatory effect of equal amounts of mitochondria and EVs, which is helpful. Oxidized mitochondria are shown to trigger a more significant IP10 response than non-stressed mitochondria, whereas EVs derived from oxidized cells do not further increase the IP10 response. Although the data are convincing, I am still wondering about a threshold effect as the IP10 response to EVs is already high. Concerning my second point, the authors should improve the description of the method in the text which is rather confusing (line 87, Fig. 1B: 120 µg mitochondria correspond to the amount of EVs isolated from 10x10⁶ cells; 12 µg, to the amount of proteins present in EVs isolated from 10x10⁶ cells).

Response to reviewers (Reviewers' comments are in italics.)

Reviewer #1 (Remarks to the Author):

Overall, the authors' revisions responded prior concerns well with one point that still needs elucidation to enhance rigor and reproducibility. This can be addressed with minor text revisions.

Key aspects of MDV analysis from confocal micrographs are still missing. For example, in the previous version of this manuscript, the authors claimed that images were quantified with ImageJ. Now, they claim that some confocal micrographs were assessed qualitatively. In one example, they state that "MDVs were considered to be colocalized with the lysosomal marker LAMP1 if they were clearly within a LAMP1-positive vesicle" This is not a standard colocalization analysis (e.g. Pearson's correlation coefficient or Mander's overlap coefficient) Authors should strongly consider rephrasing colocalization to "association" and clearly stating in the methods that their analysis is qualitative. Additionally, it is not clear how the absence of a mitochondrial marker is defined over background. It should be stated if the fluorescence intensity of these regions was measured or if this was also a qualitative assessment. The study of mitochondria-derived vesicle biology is still a nascent field, and it is critical for reproducibility to be transparent in analytical approaches.

As requested by the Reviewer, we have changed "colocalization" for "association" and modified the methods to reflect the "qualitative" nature of the quantification. Nevertheless, we disagree with the reviewer that only measures like Pearson's or Manders' coefficient can be quantitative measures. These come with their own issues (for example the quality of the segmentation greatly affecting Manders' coefficient) and are intended to measure large scale overlap between 2 markers. They cannot be applied for cases like here where only a small portion of the total signal is affected (small vesicles positive for one mitochondrial marker but not the other) and where there is a need to measure 3 markers at once (2 mitochondrial markers to define MDVs and a lysosomal marker). Manually counting MDVs is the current best quantification method in the field.

Reviewer #2 (Remarks to the Author):

The authors have done a considerable amount of additional work that has addressed my issues and has significantly improved the manuscript.

We thank the reviewer for the constructive comments.

Reviewer #3 (Remarks to the Author):

The authors have carefully addressed my concerns and significantly improved the manuscript by providing additional experimental evidence and revising the text. The observation that EVs are largely non-inflammatory is interesting and will guide further

experiments to unravel the physiological role of EVs. Moreover, the observation that increased MDV targeting to lysosomes limits the packaging of oxidized mitochondria into EVs is intriguing and sheds new light at the interface of mitochondrial quality control and inflammation, although the sorting mechanism remains unclear.

To support their key finding, the authors now compare the inflammatory effect of equal amounts of mitochondria and EVs, which is helpful. Oxidized mitochondria are shown to trigger a more significant IP10 response than non-stressed mitochondria, whereas EVs derived from oxidized cells do not further increase the IP10 response. Although the data are convincing, I am still wondering about a threshold effect as the IP10 response to EVs is already high. Concerning my second point, the authors should improve the description of the method in the text which is rather confusing (line 87, Fig. 1B: 120 μ g mitochondria correspond to the amount of EVs isolated from 10×10^6 cells; 12 μ g, to the amount of proteins present in EVs isolated from 10×10^6 cells).

Concerning the first point (the presence of a thresholding effect), this is clearly not the case for the OPA1 KO EVs, as they trigger less IP10 secretion than WT EVs (Fig. 6b). This is also consistent with our measurements of Rsad2 and mlfit1 (Fig. 6c).

Concerning the second point, we modified the text to make it clearer.